# Improving the Spatial Prediction of Soil Organic Carbon Content in Two Contrasting Climatic Regions by Stacking Machine Learning Models and Rescanning Covariate Space

**Ruhollah Taghizadeh-Mehrjardi** [1,2] , **Karsten Schmidt** [3,4,*] , **Alireza Amirian-Chakan** [5] ,
**Tobias Rentschler** [1,6] , **Mojtaba Zeraatpisheh** [7] , **Fereydoon Sarmadian** [8] , **Roozbeh Valavi** [9] ,
**Naser Davatgar** [10] , **Thorsten Behrens** [1,4,6] and **Thomas Scholten** [1,4,6]

[1] Department of Geosciences, Soil Science and Geomorphology, University of Tübingen,
72070 Tübingen, Germany; ruhollah.taghizadeh-mehrjardi@mnf.uni-tuebingen.de (R.T.-M.);
t.rentschler@uni-tuebingen.de (T.R.); thorsten.behrens@uni-tuebingen.de (T.B.);
thomas.scholten@uni-tuebingen.de (T.S.)

[2] Faculty of Agriculture and Natural Resources, Ardakan University, Ardakan 8951656767, Iran

[3] eScience Center, University of Tübingen, 72070 Tübingen, Germany

[4] DFG Cluster of Excellence "Machine Learning", University of Tübingen, 72070 Tübingen, Germany

[5] Department of Soil Science, Lorestan University, Khorramabad 6815144316, Iran; amirian.ar@lu.ac.ir

[6] CRC 1070 ResourceCultures, University of Tübingen, 72070 Tübingen, Germany

[7] Key Laboratory of Geospatial Technology for the Middle and Lower Yellow River Regions, College of
Environment and Planning, Henan University, Kaifeng 475004, China; m.zeraatpishe@alumni.iut.ac.ir

[8] Department of Soil Science, College of Agriculture, University of Tehran, Karaj 77871-31587, Iran;
fsarmad@ut.ac.ir

[9] The Quantitative & Applied Ecology Group, School of BioSciences, The University of Melbourne,
Victoria 3010, Australia; rvalavi@student.unimelb.edu.au

[10] Soil & Water Research Institute, Agricultural Research, Education and Extension Organization,
Karaj 3177993545, Iran; ndavatgar@swri.ir

[*] Correspondence: karsten.schmidt@uni-tuebingen.de

**Abstract:** Understanding the spatial distribution of soil organic carbon (SOC) content over different climatic regions will enhance our knowledge of carbon gains and losses due to climatic change. However, little is known about the SOC content in the contrasting arid and sub-humid regions of Iran, whose complex SOC–landscape relationships pose a challenge to spatial analysis. Machine learning (ML) models with a digital soil mapping framework can solve such complex relationships. Current research focusses on ensemble ML models to increase the accuracy of prediction. The usual ensemble method is boosting or weighted averaging. This study proposes a novel ensemble technique: the stacking of multiple ML models through a meta-learning model. In addition, we tested the ensemble through rescanning the covariate space to maximize the prediction accuracy. We first applied six state-of-the-art ML models (i.e., Cubist, random forests (RF), extreme gradient boosting (XGBoost), classical artificial neural network models (ANN), neural network ensemble based on model averaging (AvNNet), and deep learning neural networks (DNN)) to predict and map the spatial distribution of SOC content at six soil depth intervals for both regions. In addition, the stacking of multiple ML models through a meta-learning model with/without rescanning the covariate space were tested and applied to maximize the prediction accuracy. Out of six ML models, the DNN resulted in the best modeling accuracies, followed by RF, XGBoost, AvNNet, ANN, and Cubist. Importantly, the stacking of models indicated a significant improvement in the prediction of SOC content, especially when combined with rescanning the covariate space. For instance, the RMSE values for SOC content prediction of the upper 0–5 cm of the soil profiles of the arid site and the sub-humid site by the

---

proposed stacking approaches were 17% and 9% respectively, less than that obtained by the DNN models—the best individual model. This indicates that rescanning the original covariate space by a meta-learning model can extract more information and improve the SOC content prediction accuracy. Overall, our results suggest that the stacking of diverse sets of models could be used to more accurately estimate the spatial distribution of SOC content in different climatic regions.

**Keywords:** digital soil mapping; machine learning models; stacking of models; spatial block cross-validation; deep learning

## 1. Introduction

Soil organic carbon (SOC) storage is a key function of soils, influencing soil physicochemical properties [1,2], e.g., soil water storage capacity, nutrient holding capacity, and infiltration rate. As the world's soils contain more organic carbon than the atmosphere and the biosphere together, soils are considered to be a crucial pool in the global carbon cycle [3]. Thus, accurate information on the spatial distribution of SOC is vital to estimate and predict greenhouse gas emissions and physicochemical functions of soils [4,5]. Such information is most important in arid and semi-arid areas where soils tend to have low levels of organic carbon [6,7] compared to the humid region. These sensitive and fragile ecosystems are less resilient against climate change and, therefore, more vulnerable to desertification [8,9].

Legacy soil maps based on traditional soil mapping approaches are the most common sources for acquiring data and information on soils in Iran [9]. The qualitative nature and coarse scales of the available maps make these maps impractical for quantitative studies and a detailed understanding of the spatial variations of soil properties [10–12]. Furthermore, traditional soil mapping approaches are time-consuming and expensive [13]. Digital soil mapping approaches based on the scorpan concept [14] have become a standard approach to generate new soil data to overcome the limitations arising from the legacy soil maps. Digital soil mapping provides a quantitative-empirical framework for predicting soil properties and classes from spatially referenced covariates using appropriate machine learning (ML) models [5].

Several ML models have successfully linked SOC to environmental covariates to extrapolate SOC to unknown locations [15–29]. Some of the most popular models are multivariate regression, classical artificial neural networks [13], support vector regression [20], regression trees [17,20], and random forests [15,20,30]. Recently, deep neural networks based on deep learning approaches were used to solve highly complex soil–landscape problems [31–34]. Padarian et al. [33] and Wadoux et al. [34] predicted and mapped SOC in Chile and France respectively, using deep learning methods.

One commonly applied technique to improve the predictive capacity and to decrease the variance of the individual ML model is the ensemble model—bagging, boosting, and stacking approach [35]. Bagging is a simple and very powerful ensemble method. It generates $m$ new training sets and then $m$ models are fitted to the datasets. Their prediction results are combined by averaging the output or voting. Boosting refers to a group of algorithms that use weighted averages to turn weak learners into stronger learners. Other ensemble techniques include model averaging [36,37].

The stacking approaches combine different types of models (lower level) through a meta-learning model (higher level) to maximize the generalization accuracy [38]. Unlike bagging, boosting, and averaging methods, stacking ensemble modeling is rarely explored in digital soil mapping. Nevertheless, stacking often performs better than all individual models, especially when combined with rescanning the original covariate space [39]. For instance, Tajik et al. [40], Zhou et al. [41], and Chen et al. [42] recently evaluated the efficacy of the ensemble models—by averaging the model predictions—to predict the spatial variation of soil properties in Iran, China, and France, respectively.

Although different ML models were implemented in order to predict and map SOC [5], their performances are inconsistent in various SOC studies. To the best of our knowledge, there is no study to conduct digital mapping of SOC content using stacking approaches in different climatic conditions. Thus, the authors suggest combining the ML models with the rescanning of the original covariate space to explore if it works better than the standard stacking of individual models. Furthermore, so far, only a few studies have used deep learning models for digital soil mapping, with the notable exceptions of References [31,33,34], and a comparison with other models is still needed. This study therefore fills a void related to digital soil mapping applications of deep learning methods. Finally, there is a lack of understanding concerning the prediction of SOC content under different climatic regimes in Iran, which has a vast territory and diverse climates. Most studies conducted on SOC content in Iran only consider a single climatic influence [7,22,30].

Hence, the main objective of this study is to evaluate and compare stacking ensemble approaches with six ML models in order to predict and map the spatial distribution of SOC content for two areas with contrasting climate (i.e., arid and sub-humid) in Iran. The models include Cubist, random forests (RF), extreme gradient boosting (XGBoost), classical artificial neural network models (ANN), neural network ensemble based on model averaging (AvNNet), as well as deep learning neural networks (DNN). We further tried to identify the controlling factors of the spatial distribution patterns of SOC content in the contrasting climatic conditions, which has rarely been reported in Iran.

## 2. Materials and Methods

### 2.1. Study Sites

This study was conducted at two sites located in central (Yazd province) and northern (Gilan province) Iran (Figure 1a). The study sites comprise two diverse climatic regions (Figure 1b), which are arid in Yazd province and sub-humid in Gilan province [43]. The general climate conditions of selected sites are presented in Table 1.

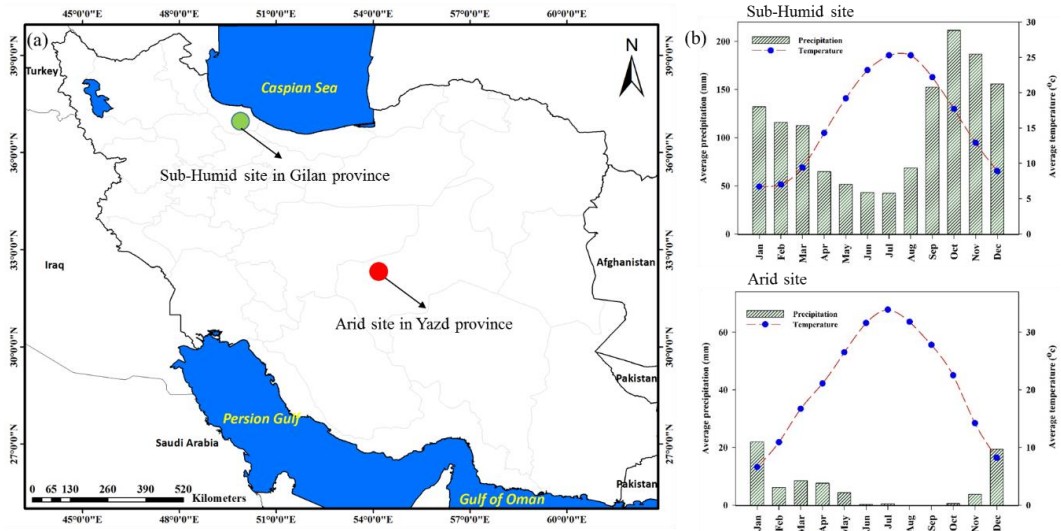

**Figure 1.** Study areas in Iran (**a**), and the ombrothermic graphs of the arid and sub-humid sites (**b**).

**Table 1.** Study sites and data collection details.

| Site Names | Area (km$^2$) | Soil Types | Climate Conditions | Precipitation (mm/year) | Elevation (m) | Samples (no.) |
|---|---|---|---|---|---|---|
| Arid site | 720 | Solonchaks, Gypsisols and Regosols | Arid | 75 | 944–1944 | 154 |
| Sub-Humid site | 3000 | Kastanozems, Cambisols and Chernozems | Sub-Humid | 1200 | −26–700 | 99 |

The arid site is located in the Yazd province in central Iran and covers 720 km$^2$. The average annual precipitation, temperature, and annual potential evaporation are 75 mm, 18.5 °C, and 3483 mm, respectively. The soil moisture and temperature regimes are aridic and thermic [43,44]. The elevation ranges from 944 to 1944 m above sea level. The main land use types consist of cropland (pistachio nuts and wheat) and grassland. The major physiographic units from East to West are alluvial fans, coalescing alluvial fans (bajadas), salt plains, and gypsiferous hills. The predominant soils in the study area [43,44] are Solonchaks with ~40%, Gypsisols with ~40%, and Regosols with ~20% of the area [43,44].

The sub-humid site is located in the Gilan province in northern Iran and covers 3000 km$^2$. The climate is sub-humid, and the average annual precipitation, temperature, and annual potential evaporation are 1200 mm, 15.6 °C, and 796 mm, respectively. The soil moisture and temperature regimes are udic and thermic [43]. The elevation ranges from −26 to 700 m above sea level. The main land use types consist of cropland (rice) and forest (oak, beech, and elm). Except for the southern parts of the study area, where piedmonts and hills dominate, the topography of the area is mainly flat. Predominant soils of the study area [43] are Kastanozems with ~70%, Cambisols with ~25%, and Chernozems with ~5% of the area.

## 2.2. Data Collection and Soil Sample Analysis

For the purpose of digital soil mapping, a well-distributed sample set is needed. We used the conditioned Latin Hypercube Sampling, which provides an optimal stratification of the covariate space [45,46], to select representative sample locations based on the covariates [47–52]. We selected a total of 154 and 99 soil profiles for the arid and sub-humid sites, respectively. Soil samples were collected from the genetic horizons of each profile down to a depth of 2 m. Air-dried soil samples were ground and sieved (<0.5 mm), and the SOC content (%) was determined by wet oxidation [53].

Sampling by genetic horizons means that samples do not come from consistent depth intervals in all locations. Therefore, we used an equal-area spline function [54] to harmonize SOC data and estimate the vertical variation of SOC content. The equal-area spline function was fitted to each profile. Then, the values of SOC content were obtained by the integration of the splines to the defined depth intervals. We estimated the SOC at six depth intervals of 0–5, 5–15, 15–30, 30–60, 60–100, and 100–200 cm, in accordance with the standard depths specified by the GlobalSoilMap project [55].

## 2.3. Covariates Used for the Development of ML Models

We used a set of 28 covariates (Table 2) as predictors [5,14] representing potential environmental drivers of the spatial and vertical distribution of SOC content. Based on the understanding of the factors affecting the SOC content distribution in the two study areas [7,22,50] and literature reviews [5,56], the covariates were obtained and derived from a digital elevation model (DEM) and remotely sensed satellite data.

The Shuttle Radar Topography Mission (SRTM) DEM with a resolution of 30 × 30 m was used for the terrain analysis [57]. The DEM was preprocessed to fill the sinks and pits before ten terrain attributes were calculated (Table 2) using SAGA GIS (System for Automated Geoscientific Analyses) [58]. These are elevation, wetness index, catchment area, catchment slope, multi-resolution valley bottom flatness index, valley depth, plane curvature, profile curvature, general curvature, and total insolation.

The remote sensing (RS)-based covariates were derived and calculated based on the median values of 127 cloud-free Landsat-8 [59] and Sentinel-2 [60] images taken during 2016 under clear and dry weather conditions during the spring/summer season using the Google Earth Engine environment [61]. In general, we used six spectral bands of Landsat-8 (B2, B3, B4, B5, B6, and B7) and ten spectral bands of Sentinel-2 (B2, B3, B4, B5, B6, B7, B8, B8a, B11, and B12), respectively [47,48]. Additionally, we calculated the normalized difference vegetation index (NDVI) using spectral bands of both Landsat-8 and Sentinel-2 [62–64].

All covariates were rescaled using Z-score standardization and resampled in order to have similar scale and the same cell size of $30 \times 30$ m.

**Table 2.** Covariates used for the development of machine learning (ML) models.

| No. | Definition | Abbreviation |
|---|---|---|
| | **Terrain-based covariates** | |
| 1 | Elevation | Elev |
| 2 | Wetness Index | WI |
| 3 | Catchments area | Ca.Area |
| 4 | Catchment Slope | Ca.Slop |
| 5 | Multi-resolution Valley Bottom Flatness Index | MrVBF |
| 6 | Valley Depth | Vally.D |
| 7 | Plane Curvature | Pl.Cur |
| 8 | Profile Curvature | Pr.Cur |
| 9 | General Curvature | Ge.Cur |
| 10 | Total Insolation | To.In |
| | **Remote Sensing-based covariates** | |
| 11 | Blue band of Landsat-8 (0.482 µm) | B2.L |
| 12 | Green band of Landsat-8 (0.561 µm) | B3.L |
| 13 | Red band of Landsat-8 (0.654 µm) | B4.L |
| 14 | Near infrared band of Landsat-8 (0.864 µm) | B5.L |
| 15 | Shortwave Infrared-1 band of Landsat-8 (1.608 µm) | B6.L |
| 16 | Shortwave Infrared-2 band of Landsat-8 (2.200 µm) | B7.L |
| 17 | Blue band of Sentinel-2 (0.490 µm) | B2.S |
| 18 | Green band of Sentinel-2 (0.560 µm) | B3.S |
| 19 | Red band of Sentinel-2 (0.665 µm) | B4.S |
| 20 | Vegetation Red Edge of Sentinel-2 (0.705 µm) | B5.S |
| 21 | Vegetation Red Edge of Sentinel-2 (0.740 µm) | B6.S |
| 22 | Vegetation Red Edge of Sentinel-2 (0.783 µm) | B7.S |
| 23 | Near infrared band of Sentinel-2 (0.842 µm) | B8.S |
| 24 | Vegetation Red Edge of Sentinel-2 (0.865 µm) | B8a.S |
| 25 | Shortwave IR-1 band of Sentinel-2 (1.610 µm) | B11.S |
| 26 | Shortwave IR-2 band of Sentinel-2 (2.190 µm) | B12.S |
| 27 | Normalized difference vegetation index (Landsat-8 based) | NDVI.L |
| 28 | Normalized difference vegetation index (Sentinel-2 based) | NDVI.S |

### 2.4. Covariate Selection

In this study, the Boruta algorithm [65] was implemented with the random forests (RF) classifier in the R statistical package [66] to rank the most important covariates for predicting SOC content at six depths. The algorithm consists of the following steps:

i.   The covariate space is extended by adding randomly permuted existing covariates (pC) in order to remove their correlation with SOC content,

ii.   A RF prediction using the extended covariate space (i.e., covariates and permuted covariates) is performed to predict SOC content at six standard depths,

iii.   The Z-score, which is an indicator of the importance of all covariates, is computed,

iv.   The maximum Z-score (MZSA) among the pC's is defined,

v.   A hit is assigned to all covariates that scored better than MZSA,

vi.   A two-test of equality is performed for undetermined important covariates,

vii.   The original covariates are respectively flagged as "unimportant" or "important" if they have significant lower or higher scores than MZSA,

viii.   All permuted covariates are removed,

ix.   Repeating the procedure.

In this study, based on Z-score values [67], we grouped the ability of covariates to explain SOC content variability into 4 classes: weakly relevant ($Z < 5$), slightly relevant ($5 < Z < 10$), moderately relevant ($10 < Z < 15$), and relevant ($Z > 15$).

## 2.5. Stacked Generalization

Stacked generalization or simply stacking is an ensemble approach that combines the outcomes of different ML models in a single model to maximize the generalization accuracy [35,37]. Usually, as illustrated in Figure 2, there are two levels in a stacking framework: level 0 and level 1, consisting of several base models and one meta-learning model. Meta-learning models in level 1 use the prediction of the response variables that are estimated by several base models in level 0 in order to generate a final prediction. In other words, the model in level 1 learns with the predictions of the models of level 0.

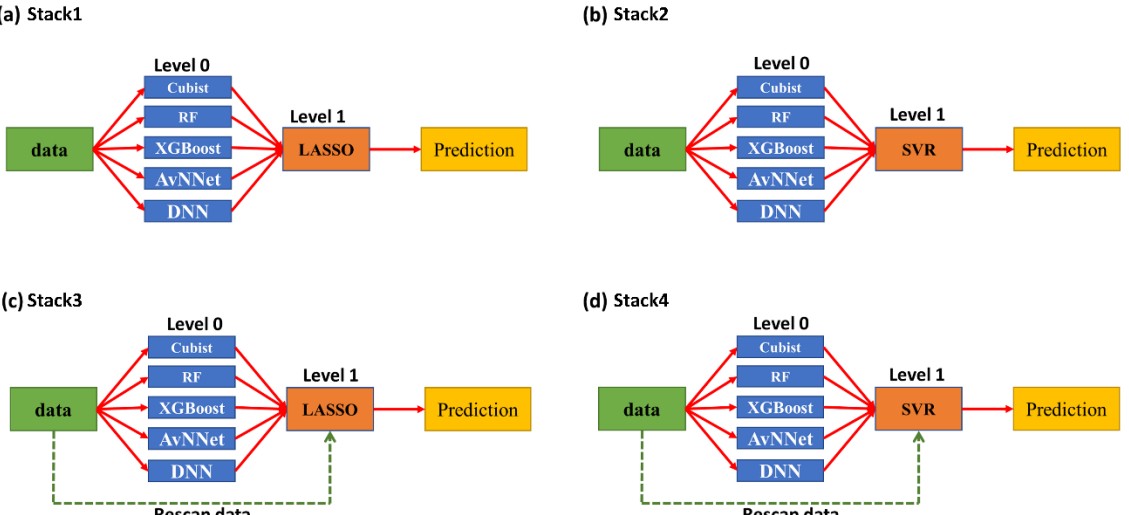

**Figure 2.** General framework of stacking approaches used in this study. (**a**) Cubist/Random Forests (RF)/extreme gradient boosting (XGBoost)/classical artificial neural network models (ANN)/neural network ensemble based on model averaging (AvNNet)/deep learning neural networks (DNN) + least absolute shrinkage and selection operator (LASSO); Stack1; (**b**) Cubist/RF/XGBoost/ANN/AvNNet/DNN + support vector regression (SVR); Stack2; (**c**) Cubist/RF/XGBoost/ANN/AvNNet/DNN + LASSO + rescan mode; Stack3; (**d**) Cubist/RF/XGBoost/ANN/AvNNet/DNN + SVR + rescan mode; Stack4.

In this study, we used six ML models (Cubist, RF, XGBoost, ANN, AvNNet, and DNN) in level 0. Conventionally, the meta-learning model in level 1 is based on a weighted average method or a linear regression model [36,37]. In this study, we applied two new meta-learning models (least absolute shrinkage and selection operator (LASSO) and support vector regression (SVR)) in level 1.

Furthermore, we introduced two modes of stacking: the standard mode (Stack1 and Stack2) and the rescan mode (Stack3 and Stack4), as shown in Figure 2 [68]. Special attention should be given to the fact that in the rescan mode, we allow the model in level 1 (LASSO and SVR) to learn again from the original covariate space in order to extract some missing information. Practically, in the standard mode, we used the predicted SOC contents of individual models (Level 0: such as RF and ANN) as the predictor variables for meta-learning models (Level 1: such as LASSO and SVR). In the rescan mode, we used both the primary covariates (such as NDVI and MrVBF) and predicted SOC contents of individual models (Level 0: such as RF and ANN) as the predictor variables for meta-learning models (Level 1: such as LASSO and SVR). In summary, we tested four stacking approaches. A more detailed account of models used in level 0 and level 1 is given in the following sections.

### 2.5.1. The Individual ML Models in Level 0

Cubist [69] is an extension of the M5 algorithm. It is similar to common regression trees, but the terminal nodes contain linear least square models of covariates used in the previous intermediate node [70] rather than discrete values. Also, there are intermediate linear least squares models at each step of the tree, which are used to adjust the final prediction. Cubist uses "if–then" rules to partition

the training data [71]. Whenever the conditions of a rule are satisfied, the associated linear least square model is used to predict the response [72].

RF [73] is an ensemble technique based on the well-known classification and regression tree approach (CART). The ensemble is generated by averaging several trees based on different bootstrap sample sets selected from the training data. Further, only a random subset of covariates is evaluated at each node. RF with a large number of trees is robust against overfitting, noise, as well as non-informative and correlated features. RF has been used in various DSM studies over the past decade [74–76] and for many other environmental problems [77]. Extreme gradient boosting (XGBoost) [78] is also a tree-based ensemble method. However, instead of independent trees and averaging the individual predictions, the XGBoost creates a number of decision trees sequentially. The trees are generated by using the residuals or prediction errors of the previous tree model, thus the algorithm focuses more on samples with higher uncertainty. Finally, all generated models are added together to calculate the outcome [79].

The most common ANNs, also known as multi-layer perceptron (MLP), consist of three layers, i.e., an input layer, a hidden layer, and an output layer. Each hidden unit combines all input units of the input layer, where all connections are associated with a weight. Further, an activation function is applied to the sum of weighted unit inputs. The output layer is calculated the same way as the hidden units, but with input from the hidden units. For the MLP with one hidden layer, we used the sigmoid function as the activation function in nnet package [80,81]. The network was trained through back-propagation using the Levenberg–Marquardt algorithm with 150 iterations [82].

AvNNet is similar to MLP, but multiple neural network models with the same topology are used to predict the response. The models can be different either due to different random number seeds to initialize the network or by fitting the models on bootstrap samples of the original training set (i.e., bagging the neural network). All the resulting models are used for prediction [81]. For regression, the outputs from each network are averaged [83]. The idea behind AvNNet is that we usually train different ANN models for the same problem in order to figure out the best model that produces the best validation statistics. However, instead of choosing the best model, it is possible to combine all models in order to improve the generalization power of a single neural network [84]. In this study, we used the AvNNet model [80,81] in the Caret package [85]. We note that the tuning parameters used for MLP were kept the same for AvNNet.

Deep learning neural networks (DNN) use the MLP structure, but have more hidden layers and a more hierarchical structure [86]. DNNs with multiple hidden layers, as shown in Appendix A (Figure A1), have a huge number of hyper-parameters (e.g., optimization algorithm, learning rate, network weight initialization, hidden layers activation function, output activation function, L2 regularization, dropout regularization, and the number of nodes in the hidden layers) [87]. The hyper-parameters potentially allow DNNs to perform better in solving the complex problems compared to the other ML models [88]. Sometimes, however, a lack of control over the learning process of the DNNs may lead to overfitting [32]. One approach, which is also used in this study to avoid or reduce overfitting, is to use a technique called Dropout [89]. Dropout randomly mutes neurons of the hidden layers. This dropout is applied to each of the $n$ training steps, resulting in $n$ different networks that are finally averaged for prediction [90]. For predictions, the ensemble of sparse networks resulting from the dropout process is averaged using the geometric mean of the input weights of the neurons. In this study, for DNNs we used the H2O package [91] with the rectifier function as a non-linear transformation and the Stochastic Gradient Descent (SGD) as the optimization algorithm. Furthermore, in order to save training time, an early stopping was used if no changes in the loss were observed after 150 epochs. Appendix A (Tables A1 and A2) shows the specifications used for DNN in this study, namely: hidden layers, size, network weight initialization, learning rate, and dropout regularization.

### 2.5.2. Meta-Learning Models in Level 1

The least absolute shrinkage and selection operator (LASSO) is a regularized linear model. It adds a regularization term as a cost function to a linear model, to reduce its degrees of freedom. To achieve

this, the lasso regression performs feature selection by eliminating the weights of the least important predictors. For the Lasso modeling, we used the glmnet package [92].

Support vector machines are a kernel method for classification [93] and regression problems [94]. The input data is transformed into a high-dimensional feature space with a predefined kernel function. In the high-dimensional feature space, a linear regression hyperplane is derived for non-linear relationships. Then, the hyperplane is back-transformed to non-linear space. The kernel used in this study is a radial basis function. The e1071 package [95] was used for radial SVR modeling.

### 2.6. Optimizing the Hyper-Parameters of Machine Learning Models

We applied a grid-learning method to estimate the best model-parameter by testing different ranges of the model parameters listed in Table 3. Importantly, these hyper-parameters are the most likely parameters to have the largest effect on the performance of the ML models. All other hyper-parameters were set to their defaults [96]. Based on the most relevant parameters, we tuned each model individually and evaluated the prediction performance. Additionally, we combined the grid-learning method with a spatial block cross-validation strategy with the aim to reduce the spatial autocorrelation effect of close neighbors and to choose the optimal model parameter. In this study, we constructed 10 folds for our block cross-validation using R package blockCV [97], in which several spatial blocks can be assigned to a fold (Figure 3). The block-to-fold assignment in this package was done by a repeated random approach that tries to find the most evenly distributed number of observations in each fold. Thus, the observations are separated spatially and in each fold as close as possible to the typical 10-fold cross-validation approach.

**Table 3.** Hyper-parameters of ML models tuned in this study.

| ML Models | Hyper-Parameters | Definition | Defined Parameters |
|---|---|---|---|
| Cubist | committees | the number of model trees | 1–100 |
| | neighbors | the number of nearest neighbors | 0–9 |
| XGboost | booster | the type of model | gbtree |
| | max_depth | the depth of tree | 3–10 |
| | min_child_weight | the minimum sum of weights of all observations | 0–5 |
| | colsample_bytree | the number of variables supplied to a tree | 0.5–1 |
| | subsample | the number of samples supplied to a tree | 0.5–1 |
| | eta | learning rate | 0.01–0.5 |
| RF | Mtry | the number of input variables | 1–30 |
| | Ntree | the number of trees | 100–3000 |
| ANN | decay | learning rate | 0.001–0.05 |
| | size | the number of neurons in hidden layer | 1–10 |
| AvNNet | Repeats | the number of MLP with different random number seeds | 3–20 |
| DNN | hidden | the number of hidden layers | 2–10 |
| | size | the number of neurons in hidden layer | 15–200 |
| | network weight initialization | the initialized weight of networks | uniform/he_normal |
| | learning rate | that controls adjusting the weights of the network | 0.001–0.05 |
| | dropout regularization | the amount of the neurons that are randomly dropped | 0.2–0.8 |
| SVM | Kernel type | the kernel function | RBF |
| | C | the penalty parameter | 0.01–100 |
| | $\sigma$ | the bandwidth parameter | 0.01–100 |
| Lasso | lambda | the shrinkage parameter | 1–150 |

MLP: multilayer perceptron; RBF: radian basis function.

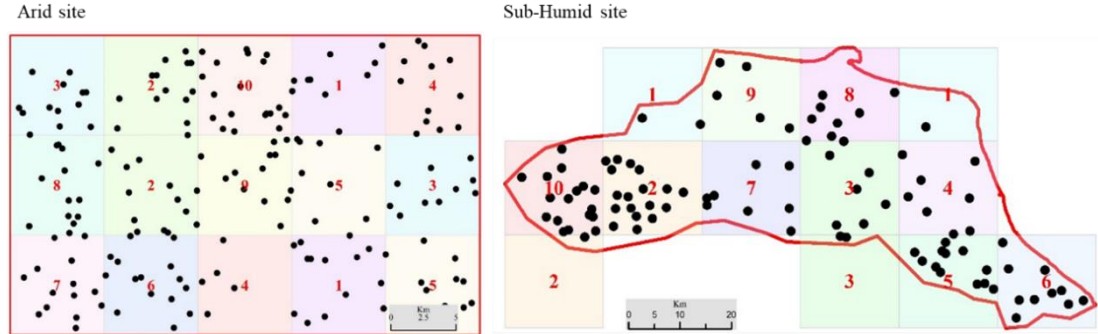

**Figure 3.** Illustration of the spatial blocking strategy in two regions. The numbers in the blocks are fold numbers, showing allocation of blocks to folds.

*2.7. Statistical Evaluation*

In this study, four common performance metrics [98], namely root mean squared error (RMSE), normalized root mean squared error (nRMSE), coefficient of determination ($R^2$), and Ratio of Performance to InterQuartile distance (RPIQ) were used. RMSE indicates the accuracy of the model prediction. nRMSE is without unit and the standardized form of RMSE and well suited for inter-model comparisons. The coefficient of determination ($R^2$) varies between 0 and 1 and indicates the closeness of the observed values to the fitted regression line or the proportion of variance explained by the independent predictors. RPIQ compares the interquartile range to the RMSE [99]. The greater the RPIQ indicates the better the model's predictive capacity.

$$RMSE = \sqrt{\frac{1}{n} \sum_{i=1}^{n} (P_i - P_o)^2} \tag{1}$$

$$nRMSE = \frac{RMSE}{\overline{O}} \tag{2}$$

$$R^2 = \left( \frac{\sum_{i=1}^{n} (O_i - \overline{O})(P_i - \overline{P})}{\sqrt{\sum_{i=1}^{n} (O_i - \overline{O})^2} \sqrt{\sum_{i=1}^{n} (P_i - \overline{P})^2}} \right)^2 \tag{3}$$

$$RPIQ = \frac{Q_3 - Q_1}{\sqrt{\frac{1}{n} \sum_{i=1}^{n} (P_i - P_o)^2}} \tag{4}$$

where Pi and Oi are the predicted and observed SOC values at the ith location, n is the number of data points, $\overline{P}$ and $\overline{O}$ denote the means for the predicted and observed SOC, and $Q_1$ and $Q_3$ are the first and third quartiles, respectively.

## 3. Results and Discussion

*3.1. Summary Statistics of SOC Content*

The descriptive statistics of the SOC content at six depth intervals across the two study areas are presented in Table 4. For the arid site, the mean SOC content varied from 0.18% to 0.33%, whereas in the sub-humid site, it ranged from 1.46% to 4.09% (Table 4). The lower and upper limits of the mean at 95% varied from 0.16 to 0.39 for the arid site, whereas in the sub-humid site, it ranged from 1.24% to 4.38%. This indicates a high variability of SOC content across the two sites. The highest variability in SOC content was found at the arid site with a coefficient of variation from 60.39% for the 60 to 100 cm depth to 128.59% for the first depth increment (0–5 cm). Similarly, the sub-humid site showed a high variability of SOC content with a coefficient of variation from 37.15% for the 0 to 5 cm depth to 78.66%

for the deepest depth increment (100–200 cm). The arid site, in contrast to the sub-humid site, tended to have higher variability in SOC content at the upmost depth increments [6,100].

**Table 4.** Descriptive statistics of soil organic carbon (SOC) content at six standard depths in two regions.

| Soil Depth | SOC (%) | | | | | | |
|---|---|---|---|---|---|---|---|
| | **Min** | **Max** | **Mean** | **Lower** | **Upper** | **SD** | **CV** |
| | **Arid site** | | | | | | |
| 0–5 cm | 0.03 | 2.34 | 0.33 | 0.26 | 0.39 | 0.42 | 128.59 |
| 5–15 cm | 0.04 | 2.21 | 0.31 | 0.25 | 0.37 | 0.39 | 124.56 |
| 15–30 cm | 0.06 | 1.69 | 0.27 | 0.23 | 0.32 | 0.30 | 110.24 |
| 30–60 cm | 0.02 | 1.11 | 0.21 | 0.19 | 0.24 | 0.17 | 77.28 |
| 60–100 cm | 0.01 | 0.75 | 0.18 | 0.16 | 0.19 | 0.11 | 60.39 |
| 100–200 cm | 0.01 | 1.00 | 0.18 | 0.16 | 0.20 | 0.14 | 78.20 |
| | **Sub-Humid site** | | | | | | |
| 0–5 cm | 1.36 | 9.93 | 4.09 | 3.79 | 4.38 | 1.52 | 37.15 |
| 5–15 cm | 1.28 | 9.51 | 3.68 | 3.41 | 3.95 | 1.39 | 37.89 |
| 15–30 cm | 0.68 | 8.01 | 2.59 | 2.34 | 2.85 | 1.30 | 50.27 |
| 30–60 cm | 0.41 | 5.65 | 1.55 | 1.35 | 1.75 | 1.03 | 66.26 |
| 60–100 cm | 0.07 | 5.65 | 1.46 | 1.24 | 1.69 | 1.15 | 78.21 |
| 100–200 cm | 0.07 | 5.65 | 1.47 | 1.24 | 1.69 | 1.15 | 78.66 |

Min: minimum; Max: maximum; SD: standard deviation; CV: coefficient of variation; Lower and Upper: the lower and upper limits of the mean at 95%.

The results of the mean SOC content comparisons for arid and sub-humid sites are shown in Figure 4. At the sub-humid site, the upper three layers (0–30 cm) are significantly different in terms of mean SOC content, whereas the lower depth intervals show no significant differences in SOC content. This result indicates more variation in the vertical distribution of SOC content at the topsoil compared to the subsoil. The mean SOC contents in the arid site show a relatively different trend compared to the sub-humid site. At the arid site, there are no significant differences between the three upper horizons. This indicated that for the top three layers (0–30 cm), depths intervals had no significant effect on SOC content.

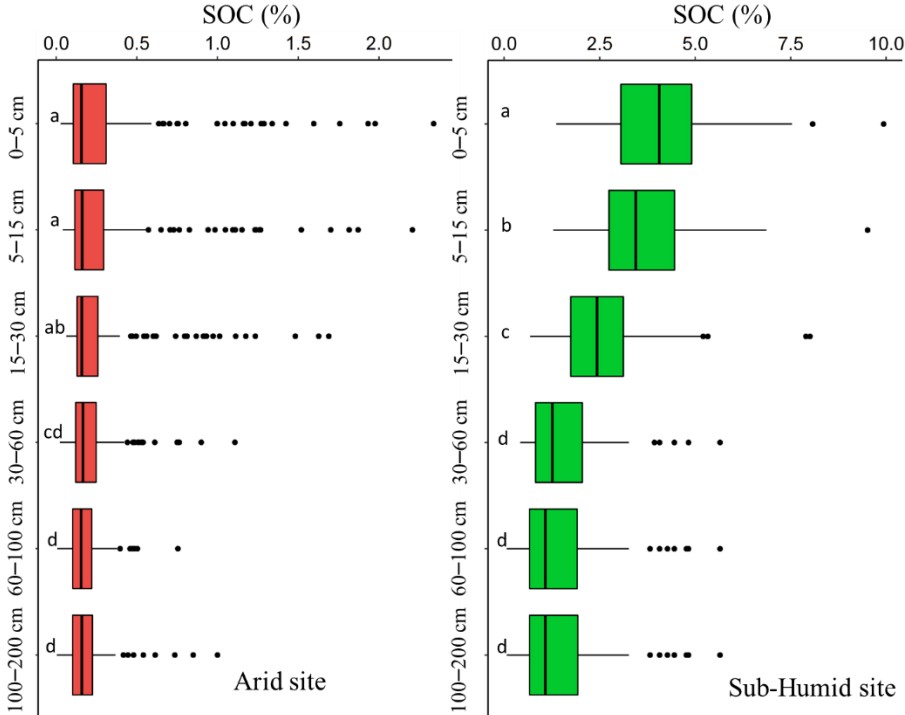

**Figure 4.** Box plot of SOC content at six standard depths at the two regions. Different letters (a:d) indicate a significant difference of mean SOC content at the 0.05 level.

A decreasing trend in SOC content with increasing depth was found on both sites. This is much more evident at the sub-humid site (Figure 4). The SOC content in both arid and sub-humid areas at the surface layer (0–5 cm) were about 1.8 and 2.8 times higher than the SOC content in the depth of 100 to 200 cm (Table 4). Several studies reported that SOC content in the topsoil was more abundant than in the subsoil [7,22,51,54,101].

### 3.2. Importance of Covariates

The selected covariates for prediction of the SOC content at the two sites at all specific depths are presented in Figure 5. The numbers indicate Z-scores and the intensity of colors from light to dark represents the values of Z-scores from low to high, respectively. The covariates used to predict SOC content showed a varying level of importance in the models. Results indicated that the covariates in the arid site were weakly to moderately relevant to SOC content. The Z-score varied from 0.40 to 10.60 for valley depth (SOC 100–200 cm) and the near infrared band of Sentinel-2 (SOC 5–15 cm), respectively. In the sub-humid site, it varied from 0.20 to 23.80 for the variables of total insolation (SOC 100–200 cm) and the Green band of Sentinel-2 (SOC 0–5 cm), respectively.

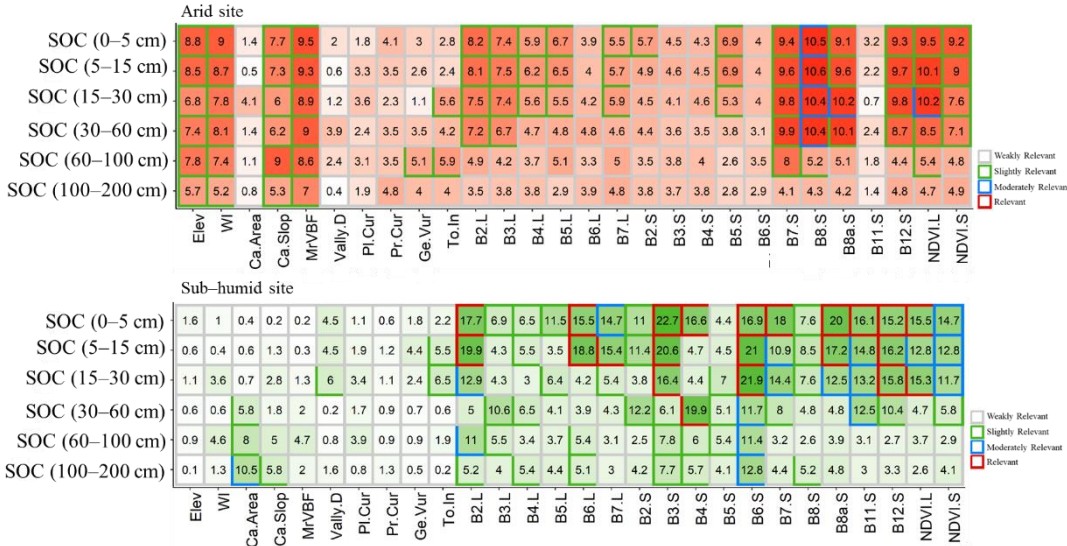

**Figure 5.** Optimal covariate selection using Boruta algorithm for SOC content at six standard depths at the two regions. The covariate is defined in Table 2.

For the arid site, Figure 5 shows that all covariates were moderately relevant (e.g., NIR band of Sentinel-2) or slightly relevant (e.g., Elevation and wetness index) to predict SOC content, at least at one soil depth interval, except for nine covariates (e.g., Catchments area). Four covariates (e.g., MrVBF) were identified as slightly relevant at all six soil depths (Figure 5). In the sub-humid site, however, only one covariate (vegetation red edge of Sentinel-2) was important at all six depth intervals and six covariates (e.g., Elevation) out of 28 were weakly relevant in predicting SOC at any depth interval (Figure 5). It should be highlighted that all other covariates were classified as slightly relevant (e.g., Catchment slope), moderately relevant (e.g., NDVI), and relevant (e.g., Green band of Sentinel-2) variables in predicting SOC content at all six depth intervals.

For the arid site, according to Figure 5, one can further conclude that both terrain- and RS-based covariates play an important role in the prediction of SOC content at six depth intervals, which is in line with the study of Wang et al. [19,102]. Importantly, the RS-based covariates (e.g., NIR band of Sentinel-2) were more important in predicting organic carbon of surface soils in comparison with the terrain-based covariates (e.g., MrVBF). This can be because most of the area was bare except for the center of the area and, thus, the remotely sensed data could represent the spectral behavior of soil at the surface [21,22]. However, terrain-based covariates were more relevant for the SOC content

prediction at the depth of 60 to 200 cm, compared to the RS-based covariates. This is particularly true for elevation (Elev), wetness index (WI), catchment slope (Ca.Slop), and multi-resolution valley bottom flatness index (MrVBF). This local topography could define the rate of soil erosion and deposition and also the amount of incoming solar radiation, which can determine SOC distribution. In addition, the influence of terrain-based covariates on predicting SOC content can be related to the other soil properties' variations as well as the existence of cultivation in the lower elevated areas (e.g., center of the area). The major factor that controls the input, decomposition, and stabilization of organic matter into the soil is the cultivation in the center of the area. The cultivation explained the amount and variation of SOC content in the study area [76]. Consequently, in the arid site, the SOC distribution may be partly characterized by topography. In line with our results, several studies [7,22,54] reported the importance of terrain-based covariates such as WI, MrVBF, slope, and elevation for the prediction of SOC at different soil depths, because they reflect important geomorphological units, e.g., deeper soil development at the valley bottom with steep slopes surrounding as a result of erosion processes.

Whereas, for the sub-humid site, the terrain-based covariates are not controlling factors on SOC content variability, with a notable exception of catchments area in the lower depth intervals (30–200 cm). These results were to be expected because the study area in the northern parts of Iran has almost flat terrain [83]. However, the finding revealed the importance of RS-based covariates on SOC content variability in the sub-humid site (Figure 5). For instance, we found NDVI as an important predictor for SOC content at surface layers of soils [47,48]. The remote-sensed vegetation parameters and NDVI are commonly considered as good indicators of primary and ecological productivity. Hence, the remote-sensed data have been effectively applied to predict SOC contents [48]. Consequently, in the sub-humid site, these results show that the vegetation has the most effective influence on topsoil SOC. Furthermore, spectral bands of Sentinel-2 had a substantial influence on the estimation of SOC contents at the 0 to 30 cm soil depth in the sub-humid site, as shown in Figure 5. This shows the high potential of spectral data in detecting SOC and its variation. Several studies [47,103] also reported a great contribution of Sentinel-2 images to predict SOC contents of soils in the Czech Republic and France. For instance, Gholizadeh et al. [48] proved the potential Sentinel-2 image to capture the variation of SOC, especially where SOC levels were relatively high. They reported that the best SOC and Sentinel-2 spectral bands' correlations were obtained from B4 and B5 followed by B11 and B12.

General speaking, RS-based covariates mostly represent surface features indicating land use/cover, parent materials, and any other features influencing spectral behaviors. Terrain-based covariates mostly represent relief parameters which contribute to erosional and depositional process as well as to water and fine particles' accumulation and redistribution. As discussed above, both terrain- and RS-based covariates can potentially explain the variation of SOC content at the two study sites (Figure 5). Nevertheless, the relative influence of covariates is distinct at the arid and sub-humid regions. According to Figure 5, our results revealed that RS-based covariates could better explain the variation of SOC content in the sub-humid site compared to the arid site. This is expected because vegetation cover by affecting land reflectance in visible and infrared [21] makes RS-based covariates promising explanatory variables to explain SOC contents and variations in the sub-humid regions [22], because a simple correlation between the photosynthetic activity of plants and a higher amount of source material (RS-based proxies) can lead to a higher accumulation of organic carbon. In other words, the effect of vegetation on SOC content might be represented by RS-based covariates. However, terrain-based covariates by controlling erosional and depositional processes were more successful in the arid site compared to the sub-humid site to explain SOC content variations. This could be due to the fact that the climate is uniform within the arid site and therefore is not a controlling factor. However, topography is a significant factor to explain SOC content variations. The least importance to terrain-based covariates in sub-humid sites is expected and attributed to the fact that the flatness in the sub-humid site minimizes the effect of topography and elevation on the soils [104]. In addition, the relative effect of covariates varies with depth, indicating that the mechanisms involved in SOC

stabilization and dynamics at different depths may be various. Basically, the cumulative effect of these variables influences the SOC contents.

### 3.3. Performances of the Individual ML Models

For both sites, the performance of six individual ML models used in level 0 in terms of $R^2$, RMSE, and RPIQ at six depth intervals was as follows: DNN > RF > XGBoost > AvNNet > ANN > Cubist (Tables 5 and 6). Our results indicated that the RF models can well predict SOC content in the two study sites. In agreement with our findings, Keskin et al. [67] reported that RF resulted in the lowest RMSE for SOC prediction compared to the other ML models because of random selection of variables during tree building and assembly. Our experience here was also similar to the conclusions achieved by Nabiollahi et al. [30], who successfully used RF to map SOC stocks at the two depth intervals (0–30 and 30–60 cm) using RS- and terrain-based covariates, and found it performed fairly good to predict SOC at two soil depths ($R^2$ = 0.70 and 0.67, respectively). However, Were et al. [20] input a wide range of the environmental covariates (e.g., soil properties, climate variables, land cover data, relief factors, and spectral indices) into RF and ANN to map SOC stocks, and showed that ANN had lower RMSE and ME values, as well as higher $R^2$ values in predicting SOC in comparison to RF models. The studies mentioned above have shown that the output of RF models varies significantly from study to study. Although, it is difficult to explain the reasons for these differences, but the differences could be because of the different extents of the study areas, topography, sampling densities, or quantity and quality of the environmental covariates used.

**Table 5.** Performances of the ML models for SOC content at six standard depths in the arid site.

| Models | $R^2$ | RMSE | RPIQ | $R^2$ | RMSE | RPIQ | $R^2$ | RMSE | RPIQ |
|---|---|---|---|---|---|---|---|---|---|
| | | 0–5 cm | | | 5–15 cm | | | 15–30 cm | |
| Cubist | 0.76 | 0.25 | 0.84 | 0.63 | 0.24 | 0.75 | 0.63 | 0.20 | 0.67 |
| XGBoost | 0.79 | 0.20 | 1.12 | 0.71 | 0.19 | 1.02 | 0.69 | 0.17 | 0.85 |
| RF | 0.80 | 0.19 | 1.18 | 0.80 | 0.19 | 1.02 | 0.72 | 0.17 | 0.85 |
| ANN | 0.75 | 0.19 | 1.05 | 0.67 | 0.19 | 0.89 | 0.65 | 0.16 | 0.78 |
| AvNNet | 0.78 | 0.20 | 1.06 | 0.69 | 0.18 | 1.01 | 0.66 | 0.17 | 0.79 |
| DNN | 0.83 | 0.17 | 1.25 | 0.80 | 0.18 | 1.07 | 0.75 | 0.16 | 0.90 |
| Stack1 | 0.83 | 0.17 | 1.25 | 0.78 | 0.18 | 1.07 | 0.74 | 0.15 | 0.92 |
| Stack2 | 0.83 | 0.17 | 1.25 | 0.81 | 0.17 | 1.09 | 0.75 | 0.14 | 0.94 |
| Stack3 | 0.86 | 0.14 | 1.30 | 0.82 | 0.13 | 1.18 | 0.77 | 0.11 | 1.07 |
| Stack4 | 0.90 | 0.14 | 1.37 | 0.85 | 0.13 | 1.20 | 0.78 | 0.10 | 1.11 |
| | | 30–60 cm | | | 60–100 cm | | | 100–200 cm | |
| Cubist | 0.49 | 0.14 | 0.92 | 0.29 | 0.13 | 0.90 | 0.17 | 0.16 | 0.78 |
| XGBoost | 0.56 | 0.14 | 1.00 | 0.33 | 0.13 | 0.99 | 0.26 | 0.16 | 0.84 |
| RF | 0.57 | 0.14 | 1.00 | 0.35 | 0.13 | 0.99 | 0.29 | 0.16 | 0.84 |
| ANN | 0.50 | 0.13 | 0.91 | 0.29 | 0.11 | 0.97 | 0.22 | 0.15 | 0.77 |
| AvNNet | 0.53 | 0.14 | 0.92 | 0.31 | 0.12 | 0.98 | 0.24 | 0.15 | 0.83 |
| DNN | 0.64 | 0.13 | 1.08 | 0.40 | 0.13 | 0.99 | 0.39 | 0.14 | 0.90 |
| Stack1 | 0.63 | 0.11 | 1.13 | 0.41 | 0.12 | 0.99 | 0.40 | 0.13 | 0.94 |
| Stack2 | 0.62 | 0.11 | 1.12 | 0.38 | 0.11 | 1.02 | 0.39 | 0.13 | 0.94 |
| Stack3 | 0.67 | 0.10 | 1.20 | 0.43 | 0.09 | 1.15 | 0.42 | 0.11 | 0.98 |
| Stack4 | 0.72 | 0.09 | 1.29 | 0.46 | 0.08 | 1.19 | 0.44 | 0.10 | 1.06 |

$R^2$: coefficient of determination; RMSE: root mean square error; RPIQ: Ratio of Performance to Interquartile distance; Stack: refers to Figure 2.

The performance of XGBoost at both sites and six depth intervals closely followed the performance of RF (Tables 5 and 6). In terms of $R^2$, RMSE, and RPIQ, it outperformed Cubist. In agreement with our findings, Tziachris et al. [104] reported the reasonable accuracy of XGBoost in comparison with RF models to predict SOC in Greece. The higher accuracy of XGBoost can be explained because its stochastic gradient, which improves the procedure, can reduce overfitting, and can improve the prediction accuracy [67]. Furthermore, XGBoost ensemble has been shown to be capable of handling noise data due to the use of a number of decision-based tree classifiers. There are several other examples of DSM experts who applied XGBoost and RF models successfully to predict soil nutrient

in Sub-Saharan Africa [105], soil properties in the United States [106], soil pH in China [107], soil properties at the global scale [108], and the depth to bedrock at the global scale [109].

**Table 6.** Performances of the ML models for SOC content at six standard depths in the sub-humid site.

| Models | $R^2$ | RMSE | RPIQ | $R^2$ | RMSE | RPIQ | $R^2$ | RMSE | RPIQ |
|---|---|---|---|---|---|---|---|---|---|
| | | 0–5 cm | | | 5–15 cm | | | 15–30 cm | |
| Cubist | 0.78 | 1.35 | 2.00 | 0.76 | 1.26 | 1.90 | 0.66 | 1.17 | 1.62 |
| XGBoost | 0.78 | 1.28 | 2.08 | 0.76 | 1.23 | 1.92 | 0.66 | 1.10 | 1.69 |
| RF | 0.78 | 1.25 | 2.11 | 0.76 | 1.18 | 1.98 | 0.66 | 1.06 | 1.73 |
| ANN | 0.78 | 1.31 | 2.04 | 0.76 | 1.25 | 1.89 | 0.65 | 1.13 | 1.65 |
| AvNNet | 0.79 | 1.30 | 2.08 | 0.77 | 1.24 | 1.93 | 0.67 | 1.12 | 1.69 |
| DNN | 0.81 | 1.26 | 2.12 | 0.79 | 1.17 | 2.02 | 0.69 | 1.05 | 1.78 |
| Stack1 | 0.83 | 1.21 | 2.16 | 0.82 | 1.17 | 2.05 | 0.73 | 1.06 | 1.78 |
| Stack2 | 0.83 | 1.20 | 2.19 | 0.82 | 1.16 | 2.04 | 0.74 | 1.03 | 1.79 |
| Stack3 | 0.84 | 1.16 | 2.25 | 0.85 | 1.13 | 2.06 | 0.74 | 1.01 | 1.81 |
| Stack4 | 0.87 | 1.15 | 2.29 | 0.86 | 1.12 | 2.10 | 0.78 | 1.01 | 1.83 |
| | | 30–60 cm | | | 60–100 cm | | | 100–200 cm | |
| Cubist | 0.52 | 0.99 | 1.46 | 0.32 | 1.19 | 1.07 | 0.23 | 1.22 | 1.11 |
| XGBoost | 0.61 | 0.95 | 1.49 | 0.36 | 1.12 | 1.12 | 0.27 | 1.15 | 1.16 |
| RF | 0.61 | 0.92 | 1.51 | 0.38 | 1.08 | 1.14 | 0.26 | 1.14 | 1.15 |
| ANN | 0.57 | 0.97 | 1.46 | 0.33 | 1.16 | 1.08 | 0.24 | 1.18 | 1.13 |
| AvNNet | 0.62 | 0.96 | 1.50 | 0.36 | 1.15 | 1.11 | 0.28 | 1.16 | 1.17 |
| DNN | 0.66 | 0.93 | 1.52 | 0.54 | 1.09 | 1.15 | 0.44 | 1.08 | 1.24 |
| Stack1 | 0.72 | 0.91 | 1.57 | 0.55 | 1.06 | 1.20 | 0.47 | 1.04 | 1.29 |
| Stack2 | 0.70 | 0.89 | 1.58 | 0.54 | 1.06 | 1.18 | 0.49 | 1.02 | 1.29 |
| Stack3 | 0.71 | 0.86 | 1.59 | 0.60 | 1.00 | 1.22 | 0.51 | 0.97 | 1.34 |
| Stack4 | 0.74 | 0.85 | 1.61 | 0.60 | 0.97 | 1.27 | 0.54 | 0.97 | 1.36 |

$R^2$: coefficient of determination; RMSE: root mean square error; RPIQ: Ratio of Performance to Interquartile distance; Stack: refers to Figure 2.

Although Cubist resulted in relatively good predictions of SOC content at the two study sites, especially at the surface layers (Tables 5 and 6), it was outperformed by RF and XGBoost. In line with our results, Zeraatpisheh et al. [21] revealed that, in terms of $R^2$ and RMSE, Cubist was outperformed by RF. Despite, the usefulness of Cubist in explaining the relationships between soil properties and covariates and in modeling SOC content, which has been reported in several studies [93], our findings showed that the model was not very competitive with the other ML models. It is not clear why Cubist failed to produce higher accuracies in comparison to the other ML models in the current research. The different findings could be related to differences in the different processes that cause the evolution and accumulation of SOC in these soils. Furthermore, this suggests that no single ML algorithm could best serve for every landscape and that multiple models should be optimized to find the most reliable prediction model. Nevertheless, we noted that the differences in the ML models' performance at both sites and at all depth intervals were rather small.

For the two areas, the performance of classical ANN at all depth intervals closely followed the performance of AvNNet (Tables 5 and 6). The higher performance of AvNNet, compared to ANN, was also reported by Baker and Ellison [84], who evaluated and compared the performance of AvNNet and ANN in order to predict water retention data. They indicated that combining ANNs improves the ability to generalize individual component ANNs. Similarly, Meyer et al. [83] reported the higher performance of AvNNet by comparing it with three ML models—RF, ANN, and SVM—for rainfall area detection and rainfall rate assignment over Germany. Although the performance of prediction slightly improves by averaging ANNs in comparison to the classical ANN, Meyer et al. [83] concluded that predictions might have been advantageous in cases where only limited data are available for training. Our result, however, is different from that of Taghizadeh-Mehrjardi et al. [7], who found superior performance for ANNs compared to RF for the three-dimensional mapping of SOC content in the western parts of Iran. It should be noted that the prediction accuracy can be affected by several parameters, such as number of field observation, type of models, the variability of soil properties, and the ability of environmental covariates to describe SOC variations.

In the arid site, the DNN was able to account for 39% to 83% of the total variation of SOC content from the lower depth interval (100 to 200 cm) to the surface layer (0–5 cm), respectively (Table 5). For the sub-humid site (Table 6), the DNN showed the best performance. It was able to account for 44% to 81% of total SOC content variation at the depths of 100–200 cm and 0–5 cm, respectively. Overall, DNN outperformed the other individual methods and produced the most accurate results, though all five ML models in the study areas proved appropriate for SOC mapping. It can be concluded that DNN provided the best outcomes in this study compared with the other individual methods and demonstrated the ML's robustness in complex data modeling and prediction. Our results are in line with other studies in the ML literature that reported the capability of the DNN model to reveal and learn the non-linear and complex patterns underlain datasets [32,110]. In soil science literature, however, the superior performance of DNN in predicting soil properties is only reported in a few studies [32,111]. For instance, Behrens et al. [31] found the most accurate results for DSM analysis using deep learning, indicating an improvement of 4–7% compared to RF. Additionally to this example, Padarian et al. [33] and Wadoux et al. [34] successfully applied a convolutional neural network (CNN) model (a well-known DNN model) to predict different soil properties (e.g., SOC) from large spectroscopic databases. Similar to our results, they also achieved a better performance by implementing a CNN compared to other individual ML models. This is mainly because of the fact that the CNN models use the contextual covariates information as input [34]. Nevertheless, our proposed DNN used the point covariates as input in this research. We can, therefore, conclude that the higher DNN output compared to the other individual ML methods can be related to the other factors, including the power of features' or attributes' extraction from raw data. DNN models create new features (sometimes also called representations of the raw data) automatically using neural networks with many hidden layers and powerful computational resources to allow them to model highly complex functions, e.g., SOC–landscape relationships.

*3.4. Performances of the Stacking Ensemble Models*

The modeling results of the two stacking approaches, namely, the standard mode (Stack1 and Stack2) and the rescan mode (Stack3 and Stack4), for the prediction of SOC content at the six depth intervals in the arid and sub-humid areas are presented in Tables 5 and 6. The modeling accuracy is as follows: Stack4 > Stack3 > Stack2 > Stack1. For instance, the RMSEs for the SOC content prediction in the depth interval of 0 to 5 cm of both sites by Stack4 models were 17% and 5% lower than the ones obtained by Stack1 (Tables 5 and 6). Similarly, Stack4 reduced the RMSEs (23% and 6%) for SOC content prediction in 100 to 200 cm of soil profiles of the arid site and the sub-humid site, compared to the Stack1 models. Generally, stacking ensemble models in rescan mode had higher accuracy than the ones in the standard mode [68]. This might be related to the fact that the meta-learning models (LASSO and SVR) used in level 1 can recapture and extract some missing information from the original covariate space. This proves the potential of using rescanning the original covariate space to improve the performance of standard stacking methods. Note that there was no attempt to test the other ML algorithms at level 0 and level 1. By changing the number and type of individual and meta-learning models, one might possibly further improve the performance of the proposed stacking method. In addition, prediction accuracy can be further improved if the model residuals' spatial correlation structure is analyzed and then added to the determinist spatial trend. However, such an analysis was beyond the scope of our study.

In order to further understand which ML models performed the best, we illustrated the performances of the stacking and individual models in Figure 6. The graph shows that the stacking ensemble modeling in both modes (standard mode and rescan mode) indicated the higher performance in comparison to the individual models. Here, we compare the performances of the best individual model (DNN) and the best stacking model (Stack4), in terms of $R^2$ and RMSE. What can be clearly seen in Figure 6 is that the Stack4 ensemble models increased $R^2$ values and decreased the RMSE values in comparison to the DNN models. For instance, the RMSEs for SOC content prediction in 0 to 5 cm of

soil profiles of the arid site and sub-humid site by Stack4 models were 17% and 9% respectively, less than that obtained by the DNN models. Similarly, Stack4 reduced the RMSE values (28% and 10%) for SOC content prediction in 100 to 200 cm of soil profiles of the arid site and sub-humid site, compared to the DNN models.

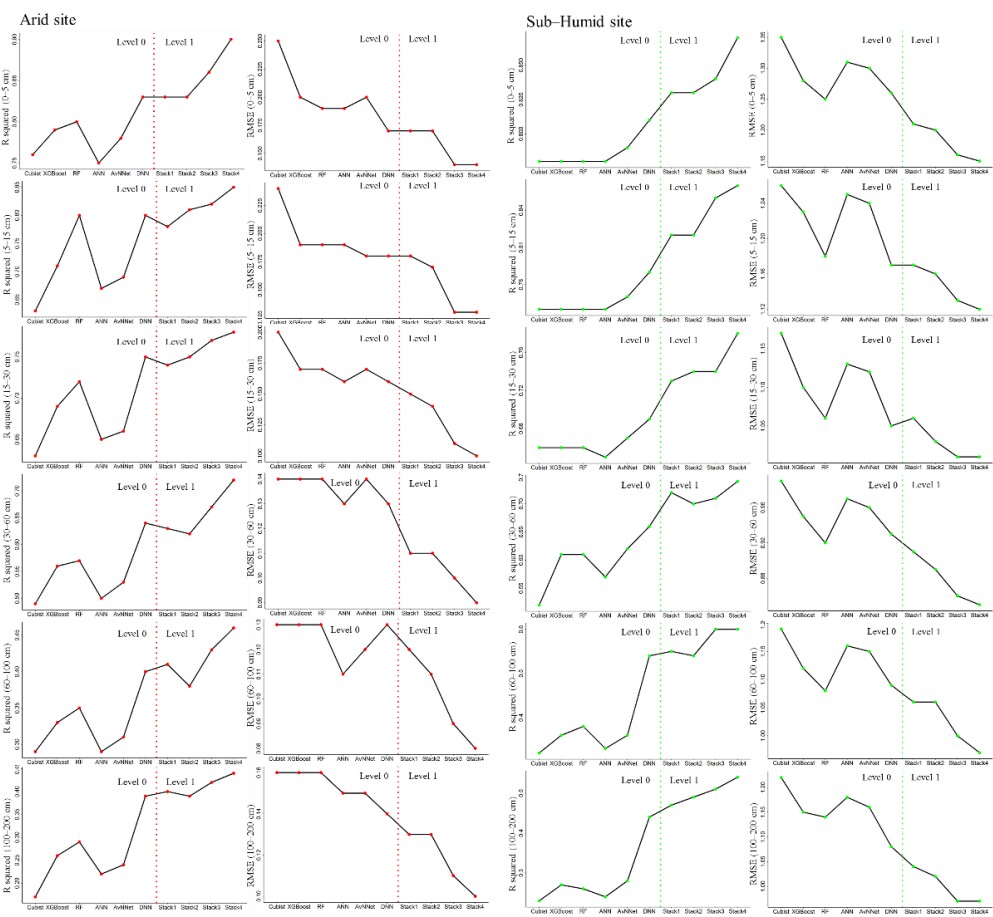

**Figure 6.** Coefficient of determination ($R^2$) and root mean squared error (RMSE) values of the individual models (Level 0) and stacking models (Level 1) for SOC content at six standard depths in two regions.

Generally, Stack4 ensemble models exhibited the best competence for capturing the spatial variation of SOC content and reducing prediction uncertainty as well. This indicated that the stacking ensemble models in level 1 were successful to keep the advantages and to discard the inaccurate aspect of the individual ML models in level 0. This is justified by the fact that the information lost by the models in level 0 is successfully captured by the level 1 models. In fact, the stacking methods used multiple learning algorithms' strengths to obtain better predictive performance and make the predictive model more robust than it is from the individual models. We emphasize that the stacking strategy is more accurate than any of its individual models if the individual models are accurate and diverse. The success of the stacking method would generally be linked to two facts: (1) the training data does not always provide enough information to select a single accurate model, and (2) the learning processes of the individual model may be imperfect [40,68]. Those are the reasons why stacking never did worse than selecting the individual models in our case study. Several studies have revealed that ensemble models exhibited the best performances for predicting soil properties in the DSM community [38,68]. For instance, Tajik et al. [38] found that stack modeling showed better performance to predict SOC content in comparison to the individual models, including RF and SVM. Similarly, Zhou et al. [39] and Chen et al. [40] recently evaluated the efficacy of the ensemble models to predict the spatial variation of soil properties. In order to predict soil total nitrogen, Zhou et al. [39] combined RF and XGBoost models

using the weighted averaging method. Their results confirmed a reasonable outcome was obtained by the ensemble approach with the lowest RMSE (1.15 g.kg$^{-1}$) and the highest R$^2$ (0.41) compared with the two individual models. Somarathna et al. [112], however, concluded that combining the ML algorithms could not provide a significant improvement (~2%) in SOC predictions. They reported that the ensemble model can only prove beneficial when combining different ML algorithms, covariates or datasets. This implies stacking strategies tend to produce reasonable performance depending on the diversification of the individual models. It can be mentioned in the current research that a diverse and powerful set of ML algorithms implemented in level 0, resulting in the stacking methods, suggested higher prediction performance compared to the individual models. Added to this, the rescanning of the original covariate space in level 1 increased the variety and strength of stacking methods, proving to be an effective tool for predicting SOC contents, which should be used as a tool in digital soil mapping.

Although the effectiveness of stacking has been demonstrated in several studies (e.g., References [38–40]), there are some potential drawbacks that should be considered. Since the most improvement in stacking is obtained when the predictors at level 0 are less correlated, selecting a combination of dissimilar models is not always an easy practice. This technique is also typically expensive in computational terms. Therefore, they add learning time and memory constraints to the problem. Last but not the least, because the stacking is a "black box" algorithm, the exact contributions of predictors to the final output cannot be explicitly disclosed. This means that the stacking (ensemble) models suffer from a lack of interpretability.

Several tools have been developed to gain insight into the fitted function and increase the interpretability of ML models, e.g., partial dependence plot (PDP) and permutation variable importance [113]. Partial plots are an intuitive and easy-to-understand visualizations technique to show the marginal effects of each predictor on the response [113,114]. However, there are limitations in using PDPs, for instance, the PDPs are meaningful when the input covariates are not highly correlated [113] and the response curve does not consider the interaction between the covariates and is more meaningful for additive models [114]. The input **feature** of stacked models (i.e., the rescanned original covariate space and the prediction of our six models) are both highly correlated (high correlation between the prediction of different models) and have a high level of interaction (as the prediction of each model directly depends on the rescanned original covariate space). Thus, using PDPs might not be very reasonable in assessing the fitted function of stacked models. This is one of the disadvantages of using model stacking as this technique makes the interpretation of models exceedingly difficult. Adding the rescan covariate space also causes further difficulties in model interpretation. In general, the interpretability of ML models in real-world problems, such as digital soil mapping, is still a challenge that needs to be focused on in the future.

## 3.5. Performances of ML Models in Two Different Climatic Regions

Vertical distribution of R$^2$, *n*RMSE, and RPIQ values to a depth of 200 cm are depicted in Figure 7. Generally, the two sites showed a decreasing trend in R$^2$ and RPIQ values with increasing depth. Otherwise, the percentage of variation in SOC content, which is described by the models, decreased with increasing depth. A reverse trend for *n*RMSE was also revealed (Figure 7). Results indicate that the models' performance decreased by each depth increment down the soil profile, and confirmed that the models had much better prediction efficiency for surface layers than subsurface layers. Increasing uncertainty of SOC content with depth has been reported in numerous studies [24]. For instance, Laub et al. [115] found that the efficiency of the ML models used for SOC prediction in China decreased from about 0.8 in the topsoil to 0.2 at 0.8 to 1 m subsoil depth. A similar pattern of uncertainty variation with depth was reported in several other studies [22]. This decreasing trend in performances could be explained by the fact that most of terrain- and RS-based covariates that are used as predictors of SOC content (listed in Table 2) explain soil surface features and processes [9] and, for example, vertical hydrological processes or bioturbation are less explained; therefore, the covariates used cannot capture the subsurface SOC variation.

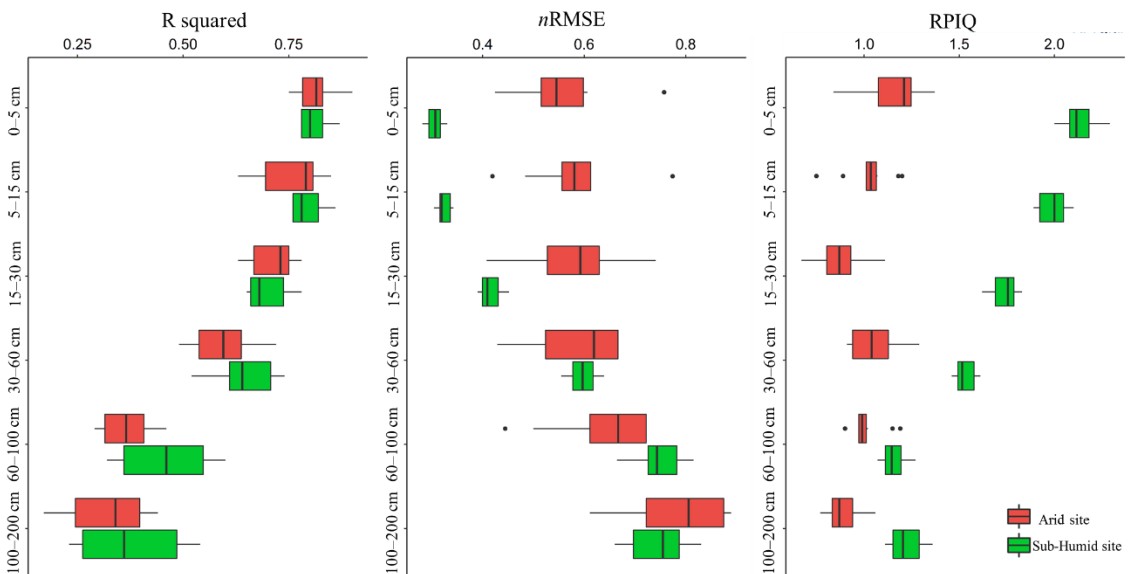

**Figure 7.** Comparison of prediction power of ML models for SOC content at six standard depths in two regions.

In line with all performance measures in Figure 7 (R$^2$, *n*RMSE, and RPIQ), all tested ML models performed better for the six soil depths at the sub-humid site than for those of the arid site. Here, we compare the performances of the best (Stack4), the worst (Cubist), and AvNNet as a model with intermediate performance, in terms of R$^2$, *n*RMSE, and RPIQ.

Stack4 resulted in R$^2$ values on average of 0.73 and 0.69 for the sub-humid site and arid site, respectively. R$^2$ values for Cubist were 0.55 and 0.50 and for AvNNet 0.58 and 0.54 for the same areas as for Stack4. Our results, furthermore, indicated that Stack4, Cubist, and AvNNet resulted in R$^2$ values for the sub-humid site that were ~9%, ~8%, and ~5% higher than the values of the arid site. The difference in the performance of ML models is much more evident when we consider *n*RMSE values, in which *n*RMSE values obtained by Stack4, Cubist, and AvNNet for the arid site were ~25%, ~17%, and ~2% more than those values for the sub-humid site (Figure 7). These results indicated that the ML models performed better at six depth intervals in the sub-humid site in comparison to those obtained in the arid site. These results could be partly attributed to large differences between areas in terms of soil forming factors, which results in complex relationships between SOC content and covariates.

The ML models resulted in a decreasing and increasing trend in R$^2$, *n*RMSE, and RPIQ respectively, with depth in the two areas (Figure 7). As can be seen, at the top layers (0-60 cm), the arid site tended to have the highest values of *n*RMSE, but with increasing depth, the accuracy of models in terms of *n*RMSE tended to be almost the same at both sites. This further shows that ML models, such as Stack4 based on the covariates used in this study, cannot capture SOC content variability at the bottom of soil profiles [7,54]. This is consistent with the results of other researchers [7,17,22,54] who all reported that the accuracy of DSM decreased with increasing depth. This indicates that the biggest uncertainty is driven by covariate space, not by the selection of an ML algorithm, which may be due to the reduction of the explanatory power of the environmental covariates along with the increase of the depth of the soil layer. It should be added that the typical covariates used in DSM studies are RS- and terrain-based covariates [10], which can only characterize the soil properties at the earth's surface. The results of this study show that there is still work to be done to improve the model accuracy for predictions of SOC in depth. Moreover, the model accuracy could potentially be improved through the use of additional environmental covariates. We assume that an implementation of important terrain-based scale-relevant covariates representing climate-driven proxy variables (e.g., shadowing effects of high mountain ranges) or the generation of covariates representing the vertical distribution of water, will influence the prediction accuracies in deeper horizons (e.g., using the geophysical techniques).

### 3.6. Spatial Distribution of SOC

The spatial distribution of mean, upper, and lower limits of SOC contents at the arid site at six interval depths is depicted in Figure 8. A decreasing trend in SOC content down the soil profile was observed. Central parts of the area tended to have the largest amounts of SOC content, which correspond to the cultivated areas mainly under pistachio orchards and wheat. Moreover, the topography of this area is mainly flat and located downslope, which results in more accumulation of fine-textured materials and water. In the arid site, because of rainfall scarcity, irrigation is necessary to provide soil moisture for crop production. Thus, irrigated farming and topographic attributes in the central parts promote more vegetation and consequently, more organic matter is accumulated in the soil. Lower SOC content in the other parts of the area can be attributed to the higher slope degree, which makes these areas prone to erosion and to higher water discharge. Further, water scarcity in these areas is not compensated by irrigation. In line with our results, Wiesmeier et al. [52] suggested that at the small scales with similar climatic conditions, vegetation, land use, and land management have a significant influence on the level of SOC stocks. At the regional scale, climatic effects may be counterbalanced by agricultural practices (e.g., fertilization and irrigation) [49].

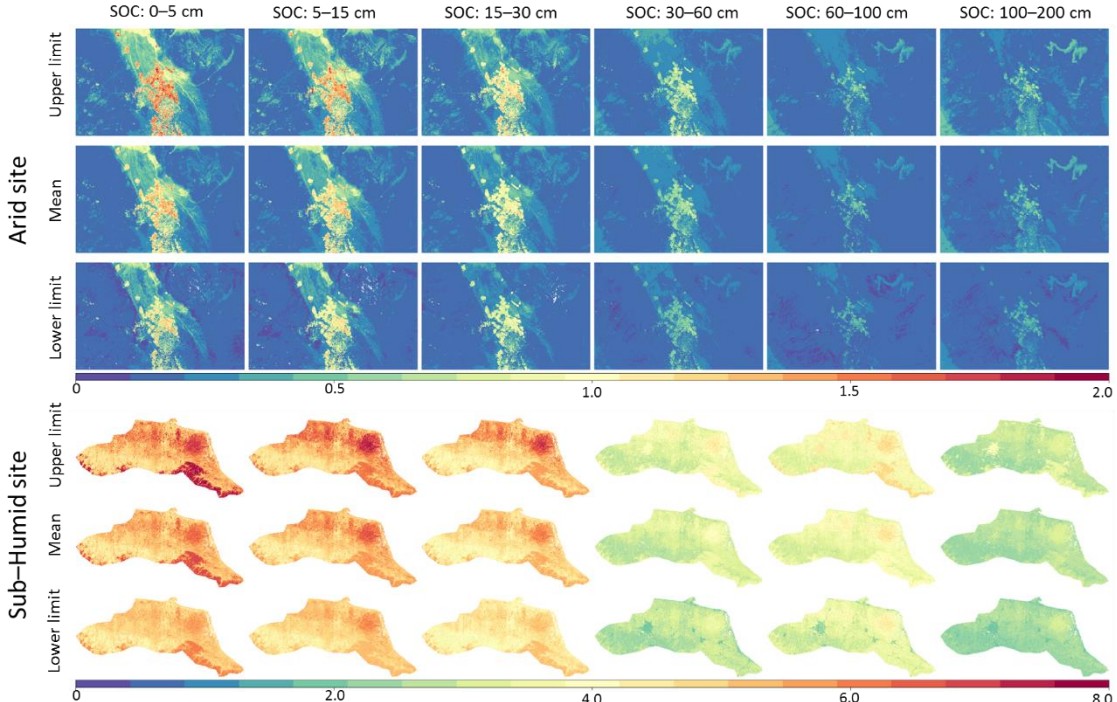

**Figure 8.** The spatial distribution of mean, upper, and lower limits of SOC contents at six standard depths in two regions. The upper and lower limits were calculated using Mean ± (1.5 × standard deviation (SD)) of the prediction values of SOC contents using the spatial block cross-validation.

The spatial distribution of mean, upper, and lower limits of SOC contents at the sub-humid site at six interval depths is depicted in Figure 8. Again, a decreasing trend in SOC content with depth is shown in the sub-humid site. The map of the spatial distribution of SOC content in the upper layer revealed more SOC accumulation in the northern parts than the other sections. The low slope degrees of the northern parts make these areas favorable for more water accumulation and, thus, result in poorly drained soils. SOC content is more accumulated and less decomposed in poorly drained soils. Mishra et al. [116] reported that high SOC stocks were found in areas characterized by low slope gradient and poorly drained soils. Wiesmeier et al. [52] indicated that areas with low slope degree and concave surface favor water accumulation. Soil moisture, which is largely controlled by terrain attributes, affects the spatial distribution of SOC content [117].

## 4. Conclusions

In this study, we introduced stacking ML models in two modes (standard mode and rescan mode) in order to improve the spatial prediction of SOC content at two contrasting climatic regions (arid and sub-humid) of Iran. The main conclusions are:

1. Though the differences in the ML models' performance at both sites and at all depth intervals were rather small, DNN was identified as the most suitable individual model.
2. The stacking ensemble modeling in both modes (standard mode and rescan mode) indicated the higher performance in comparison to the individual models.
3. Although both terrain- and RS-based covariates were important to explain SOC contents at both sites, their explanatory power was different at both sites and at the soil depth intervals.
4. The stacking models are able to explain the effect of contrasting climate on SOC content distribution. Higher content of SOC in the sub-humid site and lower content of SOC in the arid site were found, however local variation is controlled by moisture, terrain, and land use.

**Author Contributions:** Conceptualization—R.T.-M., T.B., T.S., and K.S.; methodology—R.T.-M., T.R., R.V., T.B., and K.S.; software—R.T.-M.; analysis—R.T.-M. and K.S.; investigation—R.T.-M., A.A.-C., T.B., T.S., and K.S.; data curation—R.T.-M., F.S., and N.D.; writing—original draft preparation—R.T.-M., A.A.-C., T.R., M.Z., R.V., F.S., and K.S.; visualization—R.T.-M. All authors have read and agreed to the published version of the manuscript.

**Funding:** This research received no external funding.

**Acknowledgments:** R.T.-M. has been supported by the Alexander von Humboldt Foundation (grant number: Ref3.4-1164573-IRN-GFHERMES-P). K.S., T.R., and T.S. thank the German Research Foundation (DFG) for supporting this research through the Collaborative Research Center (SFB 1070) 'ResourceCultures' (subprojects Z, S and B02). K.S., T.B., and T.S. are also supported by the DFG Cluster of Excellence "Machine Learning—New Perspectives for Science", EXC 2064/1, project number 390727645. R.V. is supported by an Australian Government Research Training Program Scholarship and a Rowden White Scholarship. Sandra Teuber, Department of Geosciences, University of Tübingen, Tübingen, Germany, thoroughly revised technical English of the paper. We acknowledge support by Open Access Publishing Fund of University of Tübingen. Finally, we thank the anonymous reviewers and editors for their careful reading of our manuscript and their many insightful comments and suggestions.

**Conflicts of Interest:** The authors declare no conflict of interest.

## Appendix A

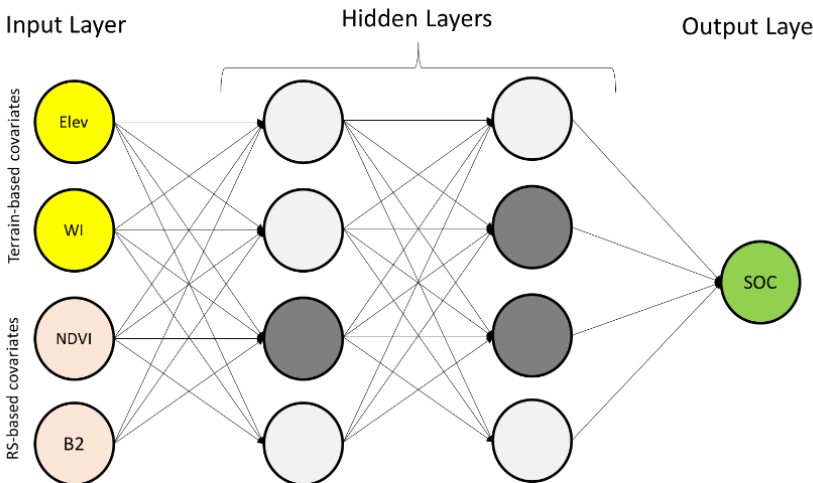

**Figure A1.** Illustration of the DNN model with two hidden layers. Black-colored circles are neurons deactivated using the dropout method. (Elev: elevation; WI: wetness index; NDVI: normalized difference vegetation index; B2: the second band of remote sensed data; SOC: soil organic carbon content).

**Table A1.** The optimal set of hyper-parameters used for ML algorithms for prediction of SOC in the arid site.

| ML Models | Hyper-Parameters | Arid Site | | | | | |
|---|---|---|---|---|---|---|---|
| | | SOC 0–5 cm | SOC 5–15 cm | SOC 15–30 cm | SOC 30–65 cm | SOC 60–100 cm | SOC 100–200 cm |
| Cubist | committees | 3 | 3 | 7 | 5 | 4 | 3 |
| | neighbors | 4 | 3 | 4 | 4 | 7 | 2 |
| XGboost | booster | gbtree | gbtree | gbtree | gbtree | gbtree | gbtree |
| | max_depth | 6 | 4 | 7 | 6 | 5 | 6 |
| | min_child_weight | 2 | 1 | 2 | 1 | 3 | 1 |
| | colsample_bytree | 0.5 | 0.5 | 0.5 | 0.5 | 0.5 | 0.5 |
| | subsample | 0.75 | 0.75 | 0.5 | 0.75 | 0.25 | 0.5 |
| | eta | 0.3 | 0.3 | 0.2 | 0.2 | 0.3 | 0.3 |
| RF | Mtry | 9 | 11 | 12 | 18 | 16 | 22 |
| | Ntree | 800 | 500 | 1100 | 1200 | 1800 | 2400 |
| ANN | decay | 0.01 | 0.01 | 0.03 | 0.03 | 0.03 | 0.01 |
| | size | 8 | 5 | 6 | 5 | 8 | 8 |
| AvNNet | Repeats | 14 | 10 | 9 | 18 | 24 | 7 |
| DNN | Hidden | 4 | 4 | 6 | 5 | 6 | 8 |
| | Size | 15 | 20 | 30 | 40 | 30 | 50 |
| | Network weight initialization | uniform | uniform | uniform | uniform | uniform | uniform |
| | learning rate | 0.02 | 0.05 | 0.01 | 0.03 | 0.01 | 0.02 |
| | dropout regularization | 0.7 | 0.6 | 0.3 | 0.4 | 0.4 | 0.8 |

**Table A2.** The optimal set of hyper-parameters used for ML algorithms for prediction of SOC in the sub-humid site.

| ML Models | Hyper-Parameters | Sub-Humid Site | | | | | |
|---|---|---|---|---|---|---|---|
| | | SOC 0–5 cm | SOC 5–15 cm | SOC 15–30 cm | SOC 30–65 cm | SOC 60–100 cm | SOC 100–200 cm |
| Cubist | Committees | 4 | 5 | 3 | 8 | 7 | 5 |
| | neighbors | 5 | 3 | 2 | 2 | 7 | 8 |
| XGboost | booster | gbtree | gbtree | gbtree | gbtree | gbtree | gbtree |
| | max_depth | 6 | 5 | 6 | 5 | 6 | 4 |
| | min_child_weight | 2 | 1 | 1 | 4 | 3 | 2 |
| | colsample_bytree | 0.5 | 0.5 | 0.5 | 0.5 | 0.5 | 0.5 |
| | subsample | 0.5 | 0.5 | 0.5 | 0.75 | 0.5 | 0.5 |
| | eta | 0.3 | 0.3 | 0.2 | 0.2 | 0.3 | 0.4 |
| RF | Mtry | 14 | 11 | 17 | 16 | 21 | 24 |
| | Ntree | 1400 | 900 | 1600 | 2100 | 2600 | 1900 |
| ANN | decay | 0.01 | 0.01 | 0.03 | 0.03 | 0.03 | 0.01 |
| | size | 8 | 5 | 6 | 5 | 8 | 8 |
| AvNNet | Repeats | 14 | 10 | 9 | 18 | 24 | 7 |
| DNN | hidden | 4 | 4 | 6 | 5 | 6 | 8 |
| | size | 50 | 20 | 40 | 40 | 50 | 60 |
| | Network weight initialization | uniform | uniform | uniform | uniform | uniform | uniform |
| | learning rate | 0.02 | 0.05 | 0.01 | 0.03 | 0.01 | 0.02 |
| | dropout regularization | 0.7 | 0.6 | 0.3 | 0.4 | 0.4 | 0.8 |

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
