# Peer review of "Improving the Spatial Prediction of Soil Organic Carbon Content in Two Contrasting Climatic Regions by Stacking Machine Learning Models and Rescanning Covariate Space"

_remotesensing, doi:10.3390/rs12071095_

Round 1

Reviewer 1 Report

Overall comment

This is another DSM paper to predict SOC based on EO sources. The potential new contribution of this submission is the methodology which creates a meta-regressor comprised of models stemming from various machine learning algorithms, to enhance the results of prediction. The proposed approach strongly based on existing approaches. In my opinion, another study case with different conditions or focusing on stacking algorithms instead of single models do not justify the novelty.

Lines 31 There is a word missing after 'can'.

Line 102-104 You have already highlighted the contrasting conditions two lines above. Please read carefully the manuscript, since there are many repetitions.

Line 110-115 Please clearly specify the overarching objective. In my view it’s not necessary to present sub-objectives, since it seems that we have a repetition of the central idea. Try to provide a clear main objective by rephrasing the lines 106-109.

Lines 148-151 It seems to me that these sentences belong to section 2.3

Lines167-168  …10 terrain attributes were calculated.., moreover, numbers from zero to ten are generally written in full text.

Lines 174-175 This is true. In these studies several pre-processing steps have performed to mask bare soil pixels. However,  it is not clear for the reader if you use as an extra covariate a temporal synthetic bare soil mosaic (at least it is not displayed in the table, so why you mention it?).

Line 353 Sentinel-2

Line 385 Figure 5, Please check the figure number

Line 434 …the DNN…

Line 444 Please specify that these authors predict soil properties utilizing large spectroscopic databases (and not RS covariates).

Lines 446-460 In general, CNNs are considered one of the most prominent representatives of the deep learning algorithms. However, it is true that comparing to other techniques, their interpretability is more difficult. In this context, they are generally described as “black box” models. However this is not absolutely true since, there are existing techniques that enable researchers to have a critical insight into the model and derive appropriate interpretations (e.g. visualization of network components). In my view, the implementation (lines 261-275) and discussion of DNN model is very poor. For instance, the structure of the proposed network to predict SOC from multiple channels is not described appropriately. The authors should consider to remove this part and focus on stacking modelling.  Maybe a critical view to the recent study of Padarian et al 2019 will be useful in order to provide a more detailed application of DNN. Otherwise we can conclude to erroneous conclusions (e.g. stacking have a better performance that DNN models)

Padarian, J.; Minasny, B.; Montazerolghaem, M.; Padarian, J.; Ferguson, R.; Bailey, S.; McBratney, A.B. Convolutional neural network for simultaneous prediction of several soil properties using visible/near-infrared, mid-infrared, and their combined spectra. Geoderma 2019, 352, 251-267.

Line 475 We illustrated the performance…

Line 538-539 This point should be highlighted and further discussed in the current manuscript.

In this context, please specify clearly the problem you perceive with stacking model. The statement you make now is not substantiated. Unfortunately, key results of the current research can be those derived by the interpretability of the L0 models (instead of the results of the Boruta algorithm). In the current version, these results were not interpreted in deep. I will suggest the authors re-write the relevant sections focus on the results presented in this step. However, this step must be done with care because the important features identified by the L1 model are as good as the importance generated by the L0 models. Similar studies have recently presented proposing novel stacked model that can enhance the predictions of the L0 models, as well as the framework proposed retains all the interpretability properties of the base models. Maybe a critical view to the recent study of Tsakiridis et al 2019 will be useful in justifying your statement.

Tsakiridis, NL, Tziolas, NV, Theocharis, JB, Zalidis, GC. A genetic algorithm‐based stacking algorithm for predicting soil organic matter from vis–NIR spectral data. Eur J Soil Sci. 2019; 70: 578– 590. https://doi.org/10.1111/ejss.12760

Moreover, potential drawbacks are not presented. For instance, is it possible to have direct interpretation of the L1 model, or we should consider that the final model is a combination of different models that could be stem from various covariates?

Author Response

This is another DSM paper to predict SOC based on EO sources. The potential new contribution of this submission is the methodology which creates a meta-regressor comprised of models stemming from various machine learning algorithms, to enhance the results of prediction. The proposed approach strongly based on existing approaches. In my opinion, another study case with different conditions or focusing on stacking algorithms instead of single models do not justify the novelty.

We thank the reviewer for pointing out this issue.

Iran comprises of multiple climatic regimes including arid, semi-arid, humid and sub-humid regions with the majority of arid and semi-arid regions. In most parts of the country due to limitations of soil data, there is a lack of regional and national SOC inventories. Therefore, the prediction of SOC is vital to further land use policy and decision-making. In this study, we showed that machine learning (ML) algorithm (in DSM) such as regression tree models (e.g., Cubist and Random Forest) and artificial neural network models (e.g., deep learning neural networks, DNN) could predict SOC in remote regions with lack of soil data by environmental covariates. Although several studies have been attempted to predict SOC in Iran, all of them assess DSM in the individual regions while this research tried to predict SOC in different climatic regions. In addition, most of comparison studies are based on evaluating models with a single random split of the data (e.g. 70% for training and 30% for testing) or typical random cross-validation. The comparison results provided with these approaches are overly optimistic in favor of overfitted models due to the presence of spatial autocorrelation in spatial data. We, however, implemented a block cross-validation scheme that is more robust for evaluating spatial predictions. In a conclusion, this study can be an initial step to compare the capability and performance of multiple ML to predict SOC in arid and sub-humid regions.

Furthermore, we proposed a new strategy for combining the ML models with rescanning the original input training data, and explore if it works better than the standard stacking of individual models. It should be noted that unlike bagging, boosting and averaging methods, stacking ensemble modelling is still rarely explored in digital soil mapping. Nevertheless, stacking often yields better performance than all individual models, especially when combined with rescanning the original input data. We also modified the objectives of the research at the end of Introduction section.

Lines 31 There is a word missing after 'can'.

We thank the reviewer for pointing out this issue.  We added a verb.

Line 102-104 You have already highlighted the contrasting conditions two lines above. Please read carefully the manuscript, since there are many repetitions.

We appreciate this suggestion. We modified the objectives of the research at the end of Introduction section.

Line 110-115 Please clearly specify the overarching objective. In my view it’s not necessary to present sub-objectives, since it seems that we have a repetition of the central idea. Try to provide a clear main objective by rephrasing the lines 106-109.

We appreciate this suggestion. We modified the objectives of the research at the end of Introduction section.

Lines 148-151 It seems to me that these sentences belong to section 2.3

We thank the reviewer for pointing out this issue. We agreed on that and removed the part accordingly. 

Lines167-168 …10 terrain attributes were calculated.., moreover, numbers from zero to ten are generally written in full text.

We appreciate this suggestion. We have modified throughout the manuscript.

Lines 174-175 This is true. In these studies several pre-processing steps have performed to mask bare soil pixels. However, it is not clear for the reader if you use as an extra covariate a temporal synthetic bare soil mosaic (at least it is not displayed in the table, so why you mention it?).

We thank the reviewer for pointing out this issue. We agreed on that and removed the part accordingly. 

Line 353 Sentinel-2

We appreciate this suggestion. We have modified throughout the manuscript.

Line 385 Figure 5, Please check the figure number

We appreciate this suggestion. We have checked the figures and tables.

Line 434 …the DNN…

We appreciate this suggestion. We have modified throughout the manuscript.

Line 444 Please specify that these authors predict soil properties utilizing large spectroscopic databases (and not RS covariates).

Great point. We have modified the manuscript.

Lines 446-460 In general, CNNs are considered one of the most prominent representatives of the deep learning algorithms. However, it is true that comparing to other techniques, their interpretability is more difficult. In this context, they are generally described as “black box” models. However this is not absolutely true since, there are existing techniques that enable researchers to have a critical insight into the model and derive appropriate interpretations (e.g. visualization of network components). In my view, the implementation (lines 261-275) and discussion of DNN model is very poor. For instance, the structure of the proposed network to predict SOC from multiple channels is not described appropriately. The authors should consider to remove this part and focus on stacking modelling. Maybe a critical view to the recent study of Padarian et al 2019 will be useful in order to provide a more detailed application of DNN. Otherwise we can conclude to erroneous conclusions (e.g. stacking have a better performance that DNN models)

Padarian, J.; Minasny, B.; Montazerolghaem, M.; Padarian, J.; Ferguson, R.; Bailey, S.; McBratney, A.B. Convolutional neural network for simultaneous prediction of several soil properties using visible/near-infrared, mid-infrared, and their combined spectra. Geoderma 2019, 352, 251-267.

We thank the reviewer for pointing out this issue. We agreed on that and removed the part accordingly. 

Line 475 We illustrated the performance…

Great point. We have modified the manuscript.

Line 538-539 This point should be highlighted and further discussed in the current manuscript.

In this context, please specify clearly the problem you perceive with stacking model. The statement you make now is not substantiated. Unfortunately, key results of the current research can be those derived by the interpretability of the L0 models (instead of the results of the Boruta algorithm). In the current version, these results were not interpreted in deep. I will suggest the authors re-write the relevant sections focus on the results presented in this step. However, this step must be done with care because the important features identified by the L1 model are as good as the importance generated by the L0 models. Similar studies have recently presented proposing novel stacked model that can enhance the predictions of the L0 models, as well as the framework proposed retains all the interpretability properties of the base models. Maybe a critical view to the recent study of Tsakiridis et al 2019 will be useful in justifying your statement.

Tsakiridis, NL, Tziolas, NV, Theocharis, JB, Zalidis, GC. A genetic algorithm‐based stacking algorithm for predicting soil organic matter from vis–NIR spectral data. Eur J Soil Sci. 2019; 70: 578– 590. https://doi.org/10.1111/ejss.12760

Moreover, potential drawbacks are not presented. For instance, is it possible to have direct interpretation of the L1 model, or we should consider that the final model is a combination of different models that could be stem from various covariates?

We thank the reviewer for pointing out this issue. However, we think the extending the paragraph needs to open another topic “Feature Engineering”. Therefore, we decided to remove the part accordingly. 

Reviewer 2 Report

Overall, it is an interesting piece of work in the present context of the digital mapping soil properties including organic carbon content. My major concern is that the methodology is not clear enough to be reproducible. Moreover, the paper uses some highly correlated variables such as NDVI and SAVI from both Landsat and Sentinel-2 as separate covariates. Such highly correlated variables make the assessment of the importance of them unstable. Regarding the presentation of results, presenting the 95% confidence interval of mean would be more informative than reporting CV and assessing the statistical significance of difference in mean SOC content among different soil layers would make the results more robust. Please see the following comments for more detail:

Is this paper predicting Soil organic carbon CONTENT or STOCK? If it is predicting SOC content, the word “content” should be included in the title of the article and throughout the manuscript including the objectives (line 107 – 114) and other places as relevant.

Line 116 Study sites: Please provide the geographic coordinates of the extents of your study area

Line 153: Please write an appropriate verb tense of the word ‘grind’

Line 180: You have rescaled to the range of 0 to 1 to bring the covariates to the similar ranges. How did you rescale it? Please justify why it was necessary to bring it to the similar range. Why did not you use z score standardization to bring them to a similar scale as min-max scale (0-1) may be sensitive to outliers? It may not be appropriate if you are including categorical data as well right?

Line 204 Stacked generalization: Method you used for stacking is not clear enough to reproduce the work. Did you use the predicted layers of individual models as the predictor variables for LASSO and SVR models? When you rescanned the data, did you use both the primary covariates and predicted layers as the covariates for LASSO and SVR models? I suggest that methodology be clarified in more details.

Line 336 Table 4: please provide the lower and upper limits of the mean at 95% confidence. This would be more informative than CV. I would also like to suggest you to test the significance in the difference in mean SOC among the different layers of soil.

Line 345 Importance of covariates: You have used highly correlated variables such as NDVI and SAVI from both Landsat and sentinel-2 as separate covariates. How can you expect the ranking of covariates to be stable? If the covariates are highly correlated, the machine learning techniques could take up any of the correlated variables. Although the prediction may not be affected, ranking of covariate importance is bound to be influenced. I would suggest to assess the correlation among the covariates if you want to have a robust ranking of covariates and remove one variable from the pairs of highly correlated variables prior to selecting covariates using Baruta algorithm.

Author Response

Overall, it is an interesting piece of work in the present context of the digital mapping soil properties including organic carbon content. My major concern is that the methodology is not clear enough to be reproducible. Moreover, the paper uses some highly correlated variables such as NDVI and SAVI from both Landsat and Sentinel-2 as separate covariates. Such highly correlated variables make the assessment of the importance of them unstable. Regarding the presentation of results, presenting the 95% confidence interval of mean would be more informative than reporting CV and assessing the statistical significance of difference in mean SOC content among different soil layers would make the results more robust. Please see the following comments for more detail:

Thank you so much for positive feedback. We agree with the reviewer. The description of method is modified (Lines 210-2015). We removed the SAVI and rescaled the covariates using z score and then run the models again (Figure 6, Tables 2, 5, and 6). We also calculated the 95% confidence interval of mean of SOC (Table 4). We further calculated the 95% confidence interval of mean of predicted SOC (Figure 9). Moreover, we compared the mean of SOC in different depths (Figure 4).

Is this paper predicting Soil organic carbon CONTENT or STOCK? If it is predicting SOC content, the word “content” should be included in the title of the article and throughout the manuscript including the objectives (line 107 – 114) and other places as relevant.

We appreciate this suggestion. We have now used the term of “SOC content” throughout the manuscript.

Line 116 Study sites: Please provide the geographic coordinates of the extents of your study area

Great point. We modified the figure 1.

Line 153: Please write an appropriate verb tense of the word ‘grind’

We appreciate this suggestion. We modified to “ground”.

Line 180: You have rescaled to the range of 0 to 1 to bring the covariates to the similar ranges. How did you rescale it? Please justify why it was necessary to bring it to the similar range. Why did not you use z score standardization to bring them to a similar scale as min-max scale (0-1) may be sensitive to outliers? It may not be appropriate if you are including categorical data as well right?

We thank the reviewer for pointing out this issue. We recalculated the covariates using z score standardization and rerun the models (Figure 6, Tables 2, 5, and 6). Regarding to the categorical covariates, there are not the fine scale categorical data available for both study areas. Added to this, remote sensed data can be play an important role to represent some of those categorical covariates such as land use.

Line 204 Stacked generalization: Method you used for stacking is not clear enough to reproduce the work. Did you use the predicted layers of individual models as the predictor variables for LASSO and SVR models? When you rescanned the data, did you use both the primary covariates and predicted layers as the covariates for LASSO and SVR models? I suggest that methodology be clarified in more details.

We thank the reviewer for pointing out this issue. The description of method is modified (Lines 210-2015).

Line 336 Table 4: please provide the lower and upper limits of the mean at 95% confidence. This would be more informative than CV. I would also like to suggest you to test the significance in the difference in mean SOC among the different layers of soil.

We appreciate this suggestion. We calculated the 95% confidence interval of mean of SOC (Table 4). We further calculated the 95% confidence interval of mean of predicted SOC (Figure 9).

Line 345 Importance of covariates: You have used highly correlated variables such as NDVI and SAVI from both Landsat and sentinel-2 as separate covariates. How can you expect the ranking of covariates to be stable? If the covariates are highly correlated, the machine learning techniques could take up any of the correlated variables. Although the prediction may not be affected, ranking of covariate importance is bound to be influenced. I would suggest to assess the correlation among the covariates if you want to have a robust ranking of covariates and remove one variable from the pairs of highly correlated variables prior to selecting covariates using Baruta algorithm.

Great point. We removed the SAVI and retained NDVI and then run the models again (Figure 6, Tables 2, 5, and 6).

Round 2

Reviewer 1 Report

Overall comment

The authors tried to improve the manuscript, but after the revision only my minor comments were addressed. (e.g. typos, missing words etc.). I still think that the study could be more critical, and that is why I had asked in the first round to better explain the DNN architecture and focus on interpretability of stacking model (discussions and further points not provided). I am not satisfied with their reply in the current version of the manuscript. It would have been very useful if the authors could:

  • Further provide discussion points why the results may appear moderate/better with regard to other studies presented, and not provide a simple comparison of performance metrics;
  • Put more emphasis on the interpretation of the relevant features by ascribing importance to the selected variables (variable importance of each ML model) and by broadening the discussion of this part
  • Highlight potential drawbacks and limitations of the novel stacking methodology proposed and provide suggestions for future enhancements and research
  • Provide in a much shorter and focused way specific sections (e.g. section 3.1) considering the scientific content. I do not see the need for too many lines for soil data presentation at the expense of the sections where the real discussion is performed.

Concluding the paper can be considered as appropriate for this journal considering the points aforementioned. Otherwise we can consider that there is a lack of novelty in this paper, since the authors simply mixed different existing approaches without proving any advantages or recommendations.

Author Response

-Reviewer 1

The authors tried to improve the manuscript, but after the revision only my minor comments were addressed. (e.g. typos, missing words etc.). I still think that the study could be more critical, and that is why I had asked in the first round to better explain the DNN architecture and focus on interpretability of stacking model (discussions and further points not provided). I am not satisfied with their reply in the current version of the manuscript. It would have been very useful if the authors could:

We thank the reviewer for pointing out this issue. We explained the DNN architecture with more details [please see the appendix A]. Furthermore, the interpretability of stacking models was discussed [please see the section 3.4].

Further provide discussion points why the results may appear moderate/better with regard to other studies presented, and not provide a simple comparison of performance metrics;

We thank the reviewer for pointing out this issue. We explained that why the results may appear moderate/better with regard to other studies presented [please see the section 3.3].

Put more emphasis on the interpretation of the relevant features by ascribing importance to the selected variables (variable importance of each ML model) and by broadening the discussion of this part

We thank the reviewer for pointing out this issue. It is true that we can calculate the relative importance of each ML model (ten models) for each soil depth (six depth intervals) and for two study sites. However, this resulted in 120 comparisons. Therefore, we decided to just explain that why the interpretability of ML models is difficult [please see the section 3.4]. Moreover, we explained more the contrition of covariates in the section 3.2. Future research could focus on improving the variable selection procedure.

Highlight potential drawbacks and limitations of the novel stacking methodology proposed and provide suggestions for future enhancements and research

We thank the reviewer for pointing out this issue. We explained that potential drawbacks and limitations of the novel stacking methodology proposed and provide suggestions for future enhancements and research [please see the section 3.4].

Provide in a much shorter and focused way specific sections (e.g. section 3.1) considering the scientific content. I do not see the need for too many lines for soil data presentation at the expense of the sections where the real discussion is performed.

We appreciate this suggestion. We agree with the reviewer. Therefore, we removed a paragraph and a figure accordingly [please see the section 3.1]. 

Concluding the paper can be considered as appropriate for this journal considering the points aforementioned. Otherwise we can consider that there is a lack of novelty in this paper, since the authors simply mixed different existing approaches without proving any advantages or recommendations.

Thank you so much for positive feedback.

(x) English language and style are fine/minor spell check required 

We appreciate this suggestion. The English reading was improved.

Reviewer 2 Report

Overall, the authors have improved the work a lot. With further minor amendments, the manuscript could be considered for publication. As the line numbers are NOT in English, it's hard to locate corrections made by the authors. 

Figure 1 needs to be improved as follows.

Author's have included the maps showing the geographic coordinates. It's good to show the geographic coordinates in the map. But you don't need to mention "geographic coordinates of the extents" in the caption of the figure. I would suggest you to include the latitudes and longitudes of the extents of study areas while describing the study area in the text. In the figure caption, the word "overlain" may not be correct. Did you mean to say "overlaid"? 

Furthermore, the north arrow in the map is too distracting and more prominent than the actual message the map tries to convey. Please include a simple one.

In the Figure 1.c, the boundary of study sites should be clearly visible. Shape of the study area is not visible in the last map. Or, is that rectangular box your study area?

Alternatively, as the distribution of points is mentioned in Figure 3, Figure 1.c is not necessary. You can add a box around the map of Figure 1.a and show the coordinates there. You have mentioned Persian gulf on the South. Please mention the names of surrounding countries and other seas so that the location in the context of international boundaries is clear.

In Figure 1.a and b, is dark line the boundary of the study area or a district? You may need to clarify it with some legend or labels.

In addition, while stacking the models, you have used the outputs of some algorithms as the inputs for other models. While doing so, how can you ascertain which primary variables are more important? It would be important to clarify the shortcomings of this kind of stacking approach, for example, in terms of uncertainty, error propagation and stability of the importance of predictor variables.  How could these issues be compounded by combining the outputs of some models with the primary covariates for feeding into other models?

Author Response

-Reviewer 2

Overall, the authors have improved the work a lot. With further minor amendments, the manuscript could be considered for publication. As the line numbers are NOT in English, it's hard to locate corrections made by the authors. 

Thank you so much for positive feedback.

Figure 1 needs to be improved as follows.

Author's have included the maps showing the geographic coordinates. It's good to show the geographic coordinates in the map. But you don't need to mention "geographic coordinates of the extents" in the caption of the figure. I would suggest you to include the latitudes and longitudes of the extents of study areas while describing the study area in the text. In the figure caption, the word "overlain" may not be correct. Did you mean to say "overlaid"? 

Furthermore, the north arrow in the map is too distracting and more prominent than the actual message the map tries to convey. Please include a simple one.

In the Figure 1.c, the boundary of study sites should be clearly visible. Shape of the study area is not visible in the last map. Or, is that rectangular box your study area?

Alternatively, as the distribution of points is mentioned in Figure 3, Figure 1.c is not necessary. You can add a box around the map of Figure 1.a and show the coordinates there. You have mentioned Persian gulf on the South. Please mention the names of surrounding countries and other seas so that the location in the context of international boundaries is clear.

In Figure 1.a and b, is dark line the boundary of the study area or a district? You may need to clarify it with some legend or labels.

We appreciate this suggestion. We changed the Figure 1 to cover all aforementioned comments. Furthermore, we added the latitudes and longitudes of the extents of study areas while describing the study area in the text [please see Figure 1].

In addition, while stacking the models, you have used the outputs of some algorithms as the inputs for other models. While doing so, how can you ascertain which primary variables are more important? It would be important to clarify the shortcomings of this kind of stacking approach, for example, in terms of uncertainty, error propagation and stability of the importance of predictor variables.  How could these issues be compounded by combining the outputs of some models with the primary covariates for feeding into other models?

We thank the reviewer for pointing out this issue. The interpretability of stacking models was discussed. Furthermore, the drawbacks of stacking models were mentioned [please see the section 3.4].

This manuscript is a resubmission of an earlier submission. The following is a list of the peer review reports and author responses from that submission.

Round 1

Reviewer 1 Report

Dear Authors I suggest you to publish your manuscript in a conference instead of this journal because I could not find any novelty, for example you have to propose a new ML method and then compare your results with the methods listed in the current version.

Also I send you some comments which are:

1- Abstract is too long, needs to be written in brief (I think Abstarct up to 200 words).
2- Introduction section is missing many explintion of previous related works.
3- Title of subsection 2.3 (Covariates) has to be more meaningful, not just one word.

4- You have to make the comparison between methods using the same principle. We can not compare between Tree based methods and Neural network methods. 

Author Response

Reviewer 1

Dear Authors I suggest you to publish your manuscript in a conference instead of this journal because I could not find any novelty, for example you have to propose a new ML method and then compare your results with the methods listed in the current version.

Iran comprises of multiple climatic regimes including arid, semi-arid, humid and sub-humid regions with the majority of arid and semi-arid regions. In most parts of the country due to limitations of soil data, there is a lack of regional and national SOC inventories. Therefore, the prediction of SOC is vital to further land use policy and decision-making. In this study, we showed that machine learning (ML) algorithm (in DSM) such as regression tree models (e.g., Cubist and Random Forest) and artificial neural network models (e.g., deep learning neural networks, DNN) could predict SOC in remote regions with lack of soil data by environmental covariates. Although several studies have been attempted to predict SOC, all of them assess DSM in the individual regions while this research tried to predict SOC in different climatic regions. In addition, most of comparison studies are based on evaluating models with a single random split of the data (e.g. 70% for training and 30% for testing) or typical random cross-validation. The comparison results provided with these approaches are overly optimistic in favor of overfitted models due to the presence of spatial autocorrelation in spatial data. We, however, implemented a block cross-validation scheme that is more robust for evaluating spatial predictions. In a conclusion, this study can be an initial step to compare the capability and performance of multiple ML to predict SOC in arid and sub-humid regions. We also added some text at the end of Introduction section.

Abstract is too long, needs to be written in brief (I think Abstarct up to 200 words).

We thank the reviewer for pointing out this issue. We agreed on that and removed the part accordingly. 

Introduction section is missing many explintion of previous related works.

We added the relevant references.

Title of subsection 2.3 (Covariates) has to be more meaningful, not just one word.

Done, we changed it.

You have to make the comparison between methods using the same principle. We can not compare between Tree based methods and Neural network methods.

We thank the reviewer for pointing out this issue. However, we compared the ML models from two groups: tree based and neuron based. Similar works are https://doi.org/10.1371/journal.pone.0217520; https://doi.org/10.1016/j.enbuild.2017.04.038; DOI 10.1007/s40710-017-0248-5.

Added to this, we compared all six models together in sub-section 3.3.2 and 3.4.

Reviewer 2 Report

Overall

The manuscript is well-written and discusses the significant shift which DSM is currently undergoing, mainly driven by open EO data and machine/deep learning techniques. A “positive manuscript” at the start of the new year. However, the current study strongly builds on the implementation of common machine learning methods and the novel or original part of this study needs to be brought out clearly and discussed appropriately.

Comments to authors:

Introduction

In my view, a region of interest with two different climatic regimes or comparisons of common machine learning methods do not absolutely justify the novelty (line 79-85). The present study demonstrated that the Boruta can be employed to filter out the best performing predictors from a spectrum of terrain and RS covariates and presented some preliminary findings regarding the prediction of SOC at various depth intervals. This is also mentioned by the authors (lines 524-525).

In my view these findings should be highlighted in the overarching objective in order to indicate the novel parts and discussed further in the next sections.

Line 33 these regions? specify them or rephrase

Materials and Methods

Line 109 the elevation approximately ranges...

Line 117 25 or 26 Please be in agreement with values presented in table 1

Line 125 In my opinion Fig. 2 is not necessary; I understand it is to illustrate the position of the soil samples. You can remove it or included as a part of Figure 1 since, the information (exact position) is also presented in Figure 3.

Line 133 mostly derived from remote sensing (RS) data

Line 134 elevation model. We selected…

Line 159 ...we calculated the NDVI...

Line 154-157. It is not clear to me how many images are selected in this study. Did you take into account also bare soil pixels? Please provide this information. This will help the reader to understand that landsat and S2 can provide supplementary information (difference between NDVI.L and NDVI.S). Moreover, previous studies (doi:10.3390/rs10101555) indicated that a multi-temporal data mining procedure can retrieve soil surface representation and used as an additional input to enhance soil properties quantification. The importance of this is briefly mentioned in lines 324-325. Hence, soil surface representation via multitemporal analysis of RS images may be mentioned as a critical step in the discussion in order to guide the future research.

Line 165 The definition of the RS covariates maybe confuse the reader... see covariates 20-22 and 24. In my view it could be helpful to put in bracket the spectral range. In addition you can delete the last column since all covariates are resampled to 30x30m (line 162-163).

Line 207 please complete the sentence

Line 260 three common performance metrics

Line 267  P ̅, please correct in in the appropriate format

Results and Discussion

Lines 351-378 This section could be more critical. The authors simply compare their result with other studies (e.g. other properties, performance metrics etc.). More critical discussions about benefits and limitations of modelling approach would have been good. Maybe a reference to the recent work of Tsakiridis et al 2019 (doi: 10.1111/ejss.12760) will be useful in providing examples of novel approaches where better predictions are resulted when effectively combined the complementary information contained within the various model predictions, instead of using only one model.

Line 356 Nabiollahi et.al [87], successfully

Line 363 More statistical information could be given. The authors focus on measures, which assume normality - mean and standard deviation (Section 2.8 and Table 5) and degree of determination. Moreover, information on normality (e.g., skewness, kurtosis) may be given. In my view, a presentation of RPIQ values would be helpful. In addition, graphs that showcase observed and predicted soil C stocks with evaluated methods can be included.

In addition, for each model developed, the optimal set of hyperparameters can be presented instead of the range that you have already presented in Table 3 (line 254).

Line 414 model complexity etc.

Lines 418-421 The statement you make now is absolutely true (no formula to good solutions). The limitation of this study, regarding the DNN approach (need for larger datasets), should be highlighted, as well as the need to move to an era of “explainable artificial intelligence” in order to leverage the full advantages of deep learning algorithms.

Line 519 “This further shows that ML models such as DNN cannot capture SOC variability at the bottom of soil profiles”. It would be beneficial to put this into that perspective and discuss why the prediction accuracy is not particularly high in more details.

Author Response

Reviewer 2

The manuscript is well-written and discusses the significant shift which DSM is currently undergoing, mainly driven by open EO data and machine/deep learning techniques. A “positive manuscript” at the start of the new year. However, the current study strongly builds on the implementation of common machine learning methods and the novel or original part of this study needs to be brought out clearly and discussed appropriately.

We thank the reviewer for pointing out this issue. Iran comprises of multiple climatic regimes including arid, semi-arid, humid and sub-humid regions with the majority of arid and semi-arid regions. In most parts of the country due to limitations of soil data, there is a lack of regional and national SOC inventories. Therefore, the prediction of SOC is vital to further land use policy and decision-making. In this study, we showed that machine learning (ML) algorithm (in DSM) such as regression tree models (e.g., Cubist and Random Forest) and artificial neural network models (e.g., deep learning neural networks, DNN) could predict SOC in remote regions with lack of soil data by environmental covariates. Although several studies have been attempted to predict SOC, all of them assess DSM in the individual regions while this research tried to predict SOC in different climatic regions. In a conclusion, this study can be an initial step to compare the capability and performance of multiple ML to predict SOC in arid and sub-humid regions. We also added some text at the end of Introduction section.

In my view, a region of interest with two different climatic regimes or comparisons of common machine learning methods do not absolutely justify the novelty (line 79-85). The present study demonstrated that the Boruta can be employed to filter out the best performing predictors from a spectrum of terrain and RS covariates and presented some preliminary findings regarding the prediction of SOC at various depth intervals. This is also mentioned by the authors (lines 524-525).

Done, We also added some text at the end of Introduction section.

In my view these findings should be highlighted in the overarching objective in order to indicate the novel parts and discussed further in the next sections.

Done. We also added some text at the end of Introduction section.

Line 33 these regions? specify them or rephrase

Done.

Line 109 the elevation approximately ranges...

Done.

Line 117 25 or 26 Please be in agreement with values presented in table 1

Done.

Line 125 In my opinion Fig. 2 is not necessary; I understand it is to illustrate the position of the soil samples. You can remove it or included as a part of Figure 1 since, the information (exact position) is also presented in Figure 3.

Done.

Line 133 mostly derived from remote sensing (RS) data

Done.

Line 134 elevation model. We selected…

Done.

Line 159 ...we calculated the NDVI...

Done.

Line 154-157. It is not clear to me how many images are selected in this study. Did you take into account also bare soil pixels? Please provide this information. This will help the reader to understand that landsat and S2 can provide supplementary information (difference between NDVI.L and NDVI.S). Moreover, previous studies (doi:10.3390/rs10101555) indicated that a multi-temporal data mining procedure can retrieve soil surface representation and used as an additional input to enhance soil properties quantification. The importance of this is briefly mentioned in lines 324-325. Hence, soil surface representation via multitemporal analysis of RS images may be mentioned as a critical step in the discussion in order to guide the future research.

We appreciate this suggestion. We extended the description of the covariates section.

Line 165 The definition of the RS covariates maybe confuse the reader... see covariates 20-22 and 24. In my view it could be helpful to put in bracket the spectral range. In addition you can delete the last column since all covariates are resampled to 30x30m (line 162-163).

We agree with the reviewer. We edited the Table 2.

Line 207 please complete the sentence

Done.

Line 260 three common performance metrics

Done.

Line 267  P ̅, please correct in in the appropriate format

Done.

Lines 351-378 This section could be more critical. The authors simply compare their result with other studies (e.g. other properties, performance metrics etc.). More critical discussions about benefits and limitations of modelling approach would have been good. Maybe a reference to the recent work of Tsakiridis et al 2019 (doi: 10.1111/ejss.12760) will be useful in providing examples of novel approaches where better predictions are resulted when effectively combined the complementary information contained within the various model predictions, instead of using only one model.

We thank the reviewer for pointing out this issue. We added text to the section.

Line 356 Nabiollahi et.al [87], successfully

Done.

Line 363 More statistical information could be given. The authors focus on measures, which assume normality - mean and standard deviation (Section 2.8 and Table 5) and degree of determination. Moreover, information on normality (e.g., skewness, kurtosis) may be given. In my view, a presentation of RPIQ values would be helpful. In addition, graphs that showcase observed and predicted soil C stocks with evaluated methods can be included.

Done, we added to RPIQ through the manuscript.

In addition, for each model developed, the optimal set of hyperparameters can be presented instead of the range that you have already presented in Table 3 (line 254).

Done, please see Appendix A.

Line 414 model complexity etc.

Done.

Lines 418-421 The statement you make now is absolutely true (no formula to good solutions). The limitation of this study, regarding the DNN approach (need for larger datasets), should be highlighted, as well as the need to move to an era of “explainable artificial intelligence” in order to leverage the full advantages of deep learning algorithms.

We thank the reviewer for pointing out this issue. We added text to the section.

Line 519 “This further shows that ML models such as DNN cannot capture SOC variability at the bottom of soil profiles”. It would be beneficial to put this into that perspective and discuss why the prediction accuracy is not particularly high in more details.

We thank the reviewer for pointing out this issue. We added text to the section.

Round 2

Reviewer 1 Report

Dear Authors;

You have made some changes, but this manuscript still has no novelty deserve to be published in a high quality Journal like Remote Sensing.

This manuscript (Application and Comparison of Decision Tree-Based Machine Learning Methods in Landside Susceptibility Assessment at Pauri Garhwal Area, Uttarakhand, India) compared between Decision Tree-Based methods not as you think that they compared between Tree based and Neuron Based, while the other articles you have sent me were published in conference or low quality journals.

Reviewer 2 Report

The revised manuscript was improved with most issues being addressed by the authors. Further revisions are still necessary prior to publication. I strongly recommend another proper proof-reading. Below some specific issues are mentioned:

Line 109 the current research can be an initial step:

Line 112 based on the Boruta algorithm

Line 133 The elevation ranges...

Lines 521-527 Overall, I agree with these sentence, however maybe you can present a more critical point of view, since now it is a simple statement. For example, the large training time is really a limiting factor? In digital soil mapping the model construction phase usually happens offline, so the interest lies in the development of the most accurate model possible, including techniques which can aid in the interpretation of the underlying processes.

Last but not least, looking the specific objectives I raise a concern about SO3. I realize that you tried to include it as a new novel method in the current comparative study, since it is worth to evaluate the new trend. However, the section regarding the DNN is shortly presented. In this context this task requires further elaboration. In my view the section at least should present the network architecture of the proposed DNN (e.g. pooling layers, filters etc.). Maybe you can join So2 and So3 into one to avoid put specific emphasis in DNN.